# Programmed cell death in diazotrophs and the fate of organic matter in the western tropical South Pacific Ocean during the OUTPACE cruise

Dina Spungin[1], Natalia Belkin[1], Rachel A. Foster[2], Marcus Stenegren[2], Andrea Caputo[2], Mireille Pujo-Pay[3], Nathalie Leblond[4], Cécile Dupouy[4], Sophie Bonnet[5], Ilana Berman-Frank[1*]

[1.] The Mina and Everard Goodman Faculty of Life Sciences, Bar-Ilan University, Ramat-Gan, Israel.

[2.] Stockholm University, Department of Ecology, Environment and Plant Sciences. Stockholm, Sweden.

[3.] Laboratoire d'Océanographie Microbienne – UMR 7321, CNRS - Sorbonne Universités, UPMC Univ Paris 06, Observatoire Océanologique, 66650 Banyuls-sur-mer, France.

[4.] Observatoire Océanologique de Villefranche, Laboratoire d'Océanographie de Villefranche, UMR 7093, Villefranche-sur Mer, France.

[5.] Aix-Marseille Univ., Univ. Toulon, CNRS/INSU, IRD, UM 110, Mediterranean Institute of Oceanography (MIO) UM 110, 13288, Centre IRD de Noumea, New Caledonia.

*Current address: Leon H. Charney School of Marine Sciences, University of Haifa, Mt. Carmel, Haifa 3498838, Israel

*Correspondence to*: Ilana Berman-Frank (iberman2@univ.haifa.ac.il)

**Abstract**

The fate of diazotroph (N$_2$ fixers) derived carbon (C) and nitrogen (N) and their contribution to vertical export of C and N in the Western Tropical South Pacific Ocean was studied during the OUTPACE experiment (Oligotrophy to UlTra-oligotrophy PACific Experiment). Our specific objective during OUTPACE was to determine whether autocatalytic programmed cell death (PCD), occurring in some diazotrophs, is an important mechanism affecting diazotroph mortality and a factor regulating the vertical flux of organic matter, and thus the fate of the blooms. We sampled at three long duration (LD) stations of 5 days each (LDA, LDB, and LDC) where drifting sediment traps were deployed at 150, 325 and 500 m depth. LDA and LDB were characterized by high chlorophyll *a* (Chl *a*) concentrations (0.2-0.6 µg L$^{-1}$) and dominated by dense biomass of the filamentous cyanobacteria *Trichodesmium* as well as UCYN-B and diatom-diazotroph associations (*Rhizosolenia* with *Richelia*-detected by microscopy and het-1 *nifH* copies). Station LDC was located at an ultra-oligotrophic area of the South Pacific gyre with extremely low Chl *a* concentration (~ 0.02 µg L$^{-1}$) with limited biomass of diazotrophs predominantly the unicellular UCYN-B. Our measurements of biomass from LDA and LDB yielded high activities of caspase-like and metacaspase proteases that are indicative of PCD in *Trichodesmium* and other phytoplankton. Metacaspase activity, reported here for the first time from oceanic populations, was highest at the surface of both LDA and LDB, where we also obtained high concentrations of transparent exopolymeric particles (TEP). TEP were negatively correlated with dissolved inorganic phosphorus and positively coupled to both the dissolved and particulate organic carbon pools. Our results reflect the increase in TEP production under nutrient stress and its role as a source of sticky carbon facilitating aggregation and rapid vertical sinking. Evidence for bloom decline was observed at both LDA and LDB. However, the physiological status and rates of decline of the blooms differed between the stations, influencing the amount of accumulated diazotrophic organic matter and mass flux observed in the traps during our experimental time frame. At LDA sediment traps contained the greatest export of particulate matter and significant numbers of both intact and decaying *Trichodesmium,* UCYN-B, and het-1 compared to LDB where the bloom decline began only 2 days prior to leaving the station and to LDC where no evidence for bloom or bloom decline was seen. Substantiating previous findings from laboratory cultures linking PCD to carbon export in *Trichodesmium*, our results from OUTPACE indicate that nutrient limitation may induce PCD in high biomass blooms such as *Trichodesmium* or diatom-diazotroph associations. Furthermore, PCD combined with high TEP production will tend to facilitate cellular aggregation and bloom termination and will expedite vertical flux to depth.

## 1. Introduction

The efficiency of the biological pump, essential in the transfer and sequestration of carbon to the deep ocean, depends on the balance between growth (production) and death. Moreover, the manner in which marine organisms die may ultimately determine the flow of fixed organic matter within the aquatic environment and whether organic matter is incorporated into higher trophic levels, recycled within the microbial loop, or sinks out (and is exported) to depth.

$N_2$ fixing (diazotrophic) prokaryotic organisms are important contributors to the biological pump and their ability to fix atmospheric $N_2$ confers an inherent advantage in the nitrogen-limited surface waters of many oceanic regions. The oligotrophic waters of the Western Tropical South Pacific (WTSP) have been characterized with some of the highest recorded rates of $N_2$ fixation (151-700 $\mu$mol N m$^{-2}$ d$^{-1}$) (Garcia et al., 2007; Bonnet et al., 2005), and can reach up to 1200 $\mu$mol N m$^{-2}$ d$^{-1}$ (Bonnet et al., 2017). Diazotrophic communities comprised of unicellular cyanobacteria lineages (UCYN-A, B and C), diatom-diazotroph associations such as *Richelia* associated with *Rhizosolenia,* and diverse heterotrophic bacteria such as alpha and $\gamma$- protobacteria are responsible for these rates of $N_2$ fixation. The most conspicuous of all diazotrophs, and predominating in terms of biomass, is the filamentous bloom-forming cyanobacteria *Trichodesmium* forming massive surface blooms that supply ~ 60-80 Tg N yr$^{-1}$ of the 100-200 Tg N yr$^{-1}$ of the estimated marine $N_2$ fixation (Capone et al., 1997; Carpenter et al., 2004; Westberry and Siegel,2006) with a large fraction fixed in the WTSP (Dupouy et al., 2000; Dupouy et al., 2011; Barboza Tenório et al., 2018) that may, based- on NanoSIMS cell-specific measurements, contribute up to ~ 80 % of bulk $N_2$ fixation rates in the WTSP (Bonnet et al., 2018).

How *Trichodesmium* blooms form and develop has been investigated intensely while little data is found regarding the fate of blooms. *Trichodesmium* blooms often collapse within 3-5 days, with mortality rates paralleling bloom development rates (Rodier and Le Borgne, 2008; Rodier and Le Borgne, 2010; Bergman et al., 2012). Cell mortality can occur due to grazing (O'Neil, 1998), viral lysis (Hewson et al., 2004; Ohki, 1999), and/or programmed cell death (PCD) an autocatalytic genetically controlled death (Berman-Frank et al., 2004). PCD is induced in response to oxidative and nutrient stress, as has been documented in both laboratory and natural populations of *Trichodesmium* (Berman-Frank et al., 2004; Berman-Frank et al., 2007) and in other phytoplankton (Bidle, 2015). The cellular and morphological features of PCD in *Trichodesmium*, include elevated gene expression and activity of metacaspases and caspase-like proteins; hallmark protein families involved in PCD pathways in other organisms whose functions in *Trichodesmium* are currently unknown. PCD in *Trichodesmium* also displays increased production of transparent exopolymeric particles (TEP) and trichome aggregation as well as buoyancy loss via reduction in gas vesicles. This causes rapid sinking

rates that can be significant when large biomass found in oceanic blooms crashes (Bar-Zeev et al.,
2013; Berman-Frank et al., 2004).
Simulating PCD in laboratory cultures of *Trichodesmium* in 2 m water columns (Bar-Zeev et al.,
2013) led to a collapse of the *Trichodesmium* biomass and to greatly enhanced sinking of large
aggregates, reaching rates of up to ~ 200 m d$^{-1}$, that efficiently exported particulate organic carbon
(POC) and particulate organic nitrogen (PON) to the bottom of the water column. Although the
sinking rates and degree of export from this model system could not be extrapolated to the ocean, this
study mechanistically linked autocatalytic PCD and bloom collapse to quantitative C and N export
fluxes, suggesting that PCD may have an impact on the biological pump efficiency in the oceans (Bar-
Zeev et al., 2013).
We further examined this issue in the open ocean and investigated the cellular processes
mediating *Trichodesmium* mortality in a large surface bloom from the New Caledonian lagoon
(Spungin et al., 2016). Nutrient stress induced a PCD mediated crash of the *Trichodesmium* bloom.
The filaments and colonies were characterized by upregulated expression of metacaspase genes,
downregulated expression of gas-vesicle genes, enhanced TEP production, and aggregation of the
biomass (Spungin et al., 2016). Due to experimental conditions we could not measure the subsequent
export and vertical flux of the dying biomass in the open ocean. Moreover, while the existence and
role of PCD and its mediation of biogeochemical cycling of organic matter has been investigated in
*Trichodesmium*, scarce information exists about PCD and other mortality pathways of most marine
diazotrophs.
The OUTPACE (Oligotrophy to UlTra-oligotrophy PACific Experiment) cruise was conducted
from 18 February to 3 April 2015 along a west to east gradient from the oligotrophic area north of
New Caledonia to the ultraoligotrophic western South Pacific gyre (French Polynesia). The goal of
the OUTPACE experiment was to study the diazotrophic blooms and their fate within the oligotrophic
ocean in the Western Tropical South Pacific (WTSP) Ocean (Moutin et al., 2017). Our specific
objective was to determine whether PCD was an important mechanism affecting diazotroph mortality
and a factor regulating the fate of the blooms by mediation of vertical flux of organic matter. The
strategy and experimental approach of the OUTPACE transect enabled sampling at three long
duration (LD) stations of 5 days each (referred to as stations LDA, LDB, and LDC) and provided 5-
day snapshots into diazotroph physiology, dynamics, and mortality processes. We specifically probed
for the induction and operation of PCD and examined the relationship of PCD to the fate of organic
matter and vertical flux from diazotrophs by the deployment of 3 sediment traps at 150, 325 and 500
m depths.


## 2. Methods

### 2.1. Sampling site and sampling conditions

Sampling was conducted on a transect during austral summer (18 Feb-5 Apr, 2015), on board the R/V L'Atalante (Moutin et al., 2017). Samples were collected from three long duration stations (LD-A, LD-B and LD-C) where the ship remained for 5 days at each location and 15 short duration (SD1-15) stations (approximately eight hours duration). The cruise transect was divided into two geographic regions. The first region (Melanesian archipelago, MA) included SD1-12, LDA and LDB stations (160º E-178º E and 170º-175º W). The second region (subtropical gyre, GY) included SD 13-15 and LDC stations (160º W-169º W).

### 2.2. Chlorophyll *a*

Samples for determination of (Chl *a*) concentrations were collected by filtering 550 mL sea water on GF/F filters (Whatman, UK). Filters were frozen and stored in liquid nitrogen, Chl *a* was extracted in methanol and measured fluorometrically (Turner Designs Trilogy Optical kit) (Le Bouteiller et al., 1992). Satellite derived surface Chl *a* concentrations at the LD stations were used from before and after the cruise sampling at the LD stations. Satellite Chl *a* data are added as supplementary video files (Supplementary videos S1, S2, S3).

### 2.3. Caspase-like and metacaspase activities

Biomass was collected on 25 mm, 0.2 µm pore-size polycarbonate filters and resuspended in 0.6-1 mL Lauber buffer [50 mM HEPES (pH 7.3), 100 mM NaCl, 10 % sucrose, 0.1 % (3-cholamidopropyl)-dimethylammonio-1-propanesulfonate, and 10 mM dithiothreitol] and sonicated on ice (four cycles of 30 seconds each) using an ultracell disruptor (Sonic Dismembrator, Fisher Scientific, Waltham, MA, USA). Cell extracts were centrifuged (10,000 x g, 2 min, room temperature), and the supernatant was collected for caspase-like and metacaspase activity measurements. Caspase-like specific activity (normalized to total protein concentration) was determined by measuring the kinetics of cleavage for the fluorogenic caspase substrate Z-IETD-AFC (Z-Ile-Glu-Thr-Asp-AFC) at a 50 µM final concentration (using Ex 400 nm, Em 505 nm; Synergy4 BioTek, Winooski, VT, USA), as previously described in Bar-Zeev et al. (2013). Metacaspase specific activity (normalized to total protein concentration) was determined by measuring the kinetics of cleavage for the fluorogenic metacaspase substrate Ac-VRPR-AMC (Ac-Val-Arg-Pro-Arg-AMC), (Tsiatsiani et al., 2011) at a 50 µM final concentration (using Ex 380 nm, Em 460 nm; Synergy4 BioTek, Winooski, VT, USA) (Tsiatsiani et al., 2011). Relative fluorescence units were converted to protein-normalized substrate cleavage rates using AFC and AMC standards (Sigma) for caspase-like and metacaspase activities, respectively. Total protein concentrations were determined by Pierce™ BCA protein assay kit (Thermo Scientific product #23225).

## 2.4. Phosphate analysis

Seawater for phosphate ($PO_4^{3-}$, DIP) analysis was collected in 20 mL high-density polyethylene HCl-rinsed bottles and poisoned with $HgCl_2$ to a final concentration of 20 μg $L^{-1}$, stored at 4 °C until analysis. $PO_4^{3-}$ was determined by a standard colorimetric technique using a segmented flow analyzer according to Aminot and Kérouel (2007) on a SEAL Analytical AA3 HR system 20 (SEAL Analytica, Serblabo Technologies, Entraigues Sur La Sorgue, France). Quantification limit for $PO_4^{3-}$ was 0.05 μmol $L^{-1}$.

## 2.5. Particulate organic carbon (POC) and nitrogen (PON)

Samples were filtered through pre-combusted (4 h, 450 ºC) GF/F filters (Whatman GF/F, 25 mm), dried overnight at 60 ºC and stored in a desiccator until further analysis. POC and PON were determined using a CHN analyzer Perkin Elmer (Waltham, MA, USA) 2400 Series II CHNS/O Elemental Analyzer after carbonate removal from the filters using overnight fuming with concentrated HCl vapor.

## 2.6. Dissolved organic carbon (DOC) and Total organic carbon (TOC)

Samples were collected from the Niskin bottles in combusted glass bottles and were immediately filtered through precombusted (24 h, 450 ºC) glass fiber filters (Whatman GF/F, 25 mm). Filtered samples were collected into glass precombusted ampoules that where sealed immediately after filtration. Samples were acidified with orthophosphoric acid ($H_3PO_4$) and analyzed by high temperature catalytic oxidation (HTCO) (Sugimura and Suzuki, 1988; Cauwet, 1994) on a Shimadzu TOC-L analyzer. TOC was determined as POC+DOC.

## 2.7. Transparent exopolymeric particles (TEP)

Water samples (100 mL) were gently (< 150 mbar) filtered through a 0.45 µm polycarbonate filter (GE Water & Process Technologies). Filters were then stained with a solution of 0.02 % Alcian Blue (AB) and 0.06 % acetic acid (pH of 2.5), and the excess dye was removed by a quick deionized water rinse. Filters were then immersed in sulfuric acid (80 %) for 2 h, and the absorbance (787 nm) was measured spectrophotometrically (CARY 100, Varian). AB was calibrated using a purified polysaccharide gum xanthan (GX) (Passow and Alldredge, 1995). TEP concentrations (µg GX equivalents $L^{-1}$) were measured according to Passow and Alldredge (1995). To estimate the role of TEP in C cycling, the total amount of TEP-C was calculated using the TEP concentrations at each depth, and the conversion of GX equivalents to carbon applying the revised factor of 0.63 based on empirical experiments from both natural samples from different oceanic areas and phytoplankton cultures (Engel, 2004).

## 2.8. Diazotrophic abundance

The full description of DNA extraction, primer design and qPCR analyses are described in detail in this issue (Stenegren et al., 2018). Briefly, 2.5 L of water from 6-7 depths with declining surface irradiance light intensity (100, 75, 54, 36, 10, 1, and 0.1 %) were sampled and filtered onto a 25 mm diameter Supor filter (Pall Corporation, PallNorden, AB Lund Sweden) with a pore size 0.2 μm filters. Filters were stored frozen in pre-sterilized bead beater tubes (Biospec Bartlesville Ok, USA) containing 30 mL of 0.1 mm and 0.5 mm glass bead mixture. DNA was extracted from the filters using a modified protocol of the Qiagen DNAeasy plant kit (Moisander et al., 2008) and eluted in 70 μL. With the re-eluted DNA extracts ready, samples were analyzed using the qPCR instrument StepOnePlus (Applied Biosystems) and fast mode. Previously designed TaqMAN assays and oligonucleotides and standards were prepared in advance and followed according to described methods for the following cyanobacterial diazotrophs: *Trichodesmium*, UCYN-A1, UCYN-A2, UCYN-B, *Richelia* symbionts of diatoms (het-1, het-2, het-3) (Stenegren et al., 2018; Church et al., 2005; Foster et al., 2007; Moisander et al., 2010; Thompson et al., 2012)**.**

## 2.9. Microscopy

Samples for microscopy were collected in parallel from the same depth profiles for which nucleic acids were sampled as described in Stenegren et al. (2018). Briefly, 2 profiles were collected on day 1 and 3 at each LD station and immediately filtered onto a 47 mm diameter Poretics (Millipore, Merck Millipore, Solna, Sweden) membrane filter with a pore size of 5 μm using a peristaltic pump. After filtration samples were fixed with a 1 % paraformaldehyde (v/v) for 30 min. prior to storing at -20 °C. The filters were later mounted onto an oversized slide and examined under an Olympus BX60 microscope equipped with blue (460-490 nm) and green (545-580 nm) excitation wavelengths. Three areas (0.94 mm$^2$) per filter were counted separately and values were averaged. When abundances were low, the entire filter (area=1734 mm$^2$) was observed and cells enumerated. Due to poor fluorescence, only *Trichodesmium* colonies and free-filaments could be accurately enumerated by microscopy, and in addition the larger cell diameter *Trichodesmium* (*Katagynemene pelagicum*) was counted separately as these were often present (albeit at lower densities). Other cyanobacterial diazotrophs (e.g. *Crocosphaera watsonii*-like cells, the *Richelia* symbionts of diatoms were present but with poor fluorescence and could only be qualitatively noted.

## 2.10. Particulate matter from sediment traps

Particulate matter export was quantified with three PPS5 sediment traps (1 m$^2$ surface collection, Technicap, France) deployed for 5 days at 150, 325 and 500 m at each LD station. Particle export was recovered in polyethylene flasks screwed on a rotary disk which allowed flasks to be changed automatically every 24-h to obtain a daily material recovery. The flasks were previously filled with a

buffered solution of formaldehyde (final conc. 2 %) and were stored at 4 °C until analysis to prevent
degradation of the collected material. The flask corresponding to the fifth day of sampling on the
rotary disk was not filled with formaldehyde to collect 'fresh particulate matter' for further diazotroph
quantification. Exported particulate matter was weighed and analyzed on EA-IRMS (Integra2, Sercon
Ltd) to quantify exported PC and PN.

## 2.11.   Diazotroph abundance in the traps

Triplicate aliquots of 2-4 mL from the flask dedicated for diazotroph quantification were filtered

onto 0.2 µm Supor filters, flash frozen in liquid nitrogen and stored at -80 °C until analysis. Nucleic
acids were extracted from the filters as described in Moisander et al. (2008) with a 30 second
reduction in the agitation step in a Fast Prep cell disruptor (Thermo, Model FP120; Qbiogene, Inc.
Cedex, Frame) and an elution volume of 70 µl. Diazotroph abundance for *Trichodesmium* spp.,
UCYN-B, UCYN-A1, het-1, and het-2 were quantified by qPCR analyses on the *nifH* gene using
previously described oligonucleotides and assays (Foster et al., 2007; Church et al., 2005). qPCR was
conducted using a StepOnePlus system (applied Biosystems, Life Technologies, Stockholm Sweden)
with the following parameters: 50 °C for 2 min, 95 °C for 10 min, and 45 cycles of 95 °C for 15s
followed by 60 °C for 1 min. Gene copy numbers were calculated from the mean cycle threshold (Ct)
value of three replicates and the standard curve for the appropriate primer and probe set. For each
primer and probe set, duplicate standard curves were made from 10-fold dilution series ranging from
$10^8$ to 1 gene copies per reaction. The standard curves were made from linearized plasmids of the
target *nifH* or from synthesized gBLocks gene fragments (IDT technologies, Cralville, Iowa USA).
Regression analyses of the results (number of cycles=Ct) of the standard curves were analyzed in
Excel. 2 µl of 5 KDa filtered nuclease free water was used for the no template controls (NTCs). No
*nifH* copies were detected for any target in the NTC. In some samples only 1 or 2 of the 3 replicates
produced an amplification signal; these were noted as detectable but not quantifiable (dnq). A 4th
replicate was used to estimate the reaction efficiency for the *Trichodesmium* and UCYN-B targets as
previously described in Short et al., (2004). Seven and two samples were below 95 % in reaction
efficiency for *Trichodesmium* and UCYN-B, respectively. The detection limit for the qPCR assays is
1-10 copies.

## 2.12.   Statistical analyses

A Spearman correlation coefficient test was applied to examine the strength of association

between two variables and the direction of the relationship.

## 3. Results and discussion

### 3.1. Diazotrophic characteristics and abundance in the LD stations

The sampling strategy of the transect was planned so that changes in abundance and fate of diazotrophs could be followed in "long duration" (LD) stations where measurements were taken from the same water mass (and location) over 5 days and drifting sediment traps were deployed (Moutin et al., 2017). Although rates for the different parameters were obtained for 5 days, this period is still a "snapshot" in time with the processes measured influenced by preceding events also continuing after the ship departed. Specifically, production of photosynthetic biomass (as determined from satellite-derived Chl $a$) and development of surface phytoplankton blooms, including cyanobacterial diazotrophs, displayed specific characteristics for each of the LD stations. We first examined the satellite-derived surface Chl $a$ concentrations by looking at changes around the LD stations before and after our 5-day sampling at each station [daily surface Chl $a$ (mg m$^{-3}$)] (Supplementary videos S1, S2, S3).

At LDA, satellite data confirmed high concentrations of Chl $a$ indicative of intense surface blooms (~ 0.55 µg L$^{-1}$) between 8$^{th}$ of February 2015 to 19$^{th}$ of February 2015 which began to gradually decline with over 60 % Chl $a$ reduction until day 1 at the station (Supplementary video S1, Fig. 1a). By the time we reached LDA on 25.02.15 (day 1) Chl $a$ concentrations averaged ~ 0.2 µg L$^{-1}$ Chl $a$ at the surface (Fig. 1a) and remained steady for the next 5 days with Chl $a$ values of 0.2 µg L$^{-1}$ measured on day 5 (Fig. 1a). When looking for biomass at depth the DCM recorded at ~ 80 m depth was characterized by Chl $a$ concentrations increasing from 0.4 to 0.5 µg L$^{-1}$ between day 3 and 5 respectively (Fig. 1d). While the Chl $a$ values of the surface biomass decreased for approximately one week prior to our sampling at station, the Chl $a$ concentrations measured at depth increased during the corresponding time.

In contrast to LDA, the satellite data from LDB confirmed the presence of a surface bloom/s for over one month prior to our arrival at the station on 15$^{th}$ of March 2015 (day 1) (Supplementary video S2, Fig. 1b). This bloom was characterized by high surface Chl $a$ concentrations (~ 0.6 µg L$^{-1}$, Supplementary video S2) and on day 1 at the station surface Chl $a$ was 0.6 µg L$^{-1}$ (Fig. 1b). Surface Chl $a$ then decreased over the next days at the station with a 50 % reduction of Chl $a$ concentration from the sea surface (5m) on day 5 (0.4 µg L$^{-1}$), (Fig. 1e). Thus, it appears that our 5 sampling days at LDB were tracking a surface bloom that had only began to decline after day 3 and continued to decrease (~ 0.1 µg L$^{-1}$) also after we had left the station (Fig. 1b). On day 1 of sampling, the DCM at LDB was relatively shallow, at 40 m with Chl $a$ values of 0.5 µg L$^{-1}$. By day 5 the DCM had deepened to 80 m (de Verneil et al., 2017).

LDC was located in a region of extreme oligotrophy within the Cook Islands territorial waters (GY waters). This station was characterized historically (~ 4 weeks before arrival) by extremely low

Chl *a* concentrations at the surface (~ 0.02 µg L$^{-1}$, Supplementary video S3) that were an order of magnitude lower than average Chl *a* measured at LDA and LDB. These values remained low with no significant variability for the 5 days at station or later (Fig. 1f) (Supplementary video S3, Fig. 1c). Similar to the results from LDA, the DCM at LDC was found near the bottom of the photic layer at ~ 135 m, with Chl *a* concentrations about 10-fold higher than those measured at surface with ~ 0.2 µg L$^{-1}$ (Fig. 1f).

Chl *a* is an indirect proxy of photosynthetic biomass and we thus needed to ascertain who the dominant players (specifically targeting diazotrophic populations) were at each of the LD stations. Moreover, At LDA and LDB diazotrophic composition and abundance as determined by qPCR analysis were quite similar. At LDA *Trichodesmium* was the most abundant diazotroph, ranging between 6x10$^4$-1x10$^6$ *nifH* copies L$^{-1}$ in the upper water column (0-70 m). UCYN-B (genetically identical to *Crocosphaera watsonii*) co-occurred with *Trichodesmium* between 35 and 70 m, and het1 specifically identifying the diatom-diazotroph association (DDA) between the diatom *Rhizosolenia* and the heterocystous diazotroph *Richelia,* was observed only at the surface waters at 4 m. UCYN-B and het-1 abundances were relatively lower than *Trichodesmium* abundances with 2x10$^2$ *nifH* copies L$^{-1}$ and 3x10$^3$ *nifH* copies L$^{-1}$ respectively (Stenegren et al., 2018). Microscopic observations from LDA indicated that near the surface *Rhizosolenia* populations were already showing signs of decay since the silicified cell-wall frustules were broken and free filaments of *Richelia* were observed (Fig. 2e-f) (Stenegren et al., 2018). DDAs are significant N$_2$ fixers in the oligotrophic oceans. Although their abundance in the WTSP is usually low, they are common and highly abundant in the New Caledonian lagoon significantly impacting C sequestration and rapid sinking (Turk-Kubo et al., 2015).

At LDB, *Trichodesmium* was also the most abundant diazotroph with *nifH* copies L$^{-1}$ ranging between 1x10$^4$-5x10$^5$ within the top 60 m (Stenegren et al., 2018). Microscopical analyses confirmed high abundance of free filaments of *Trichodesmium* at LDB, while colonies were rarely observed (Stenegren et al., 2018). Observations of poor cell integrity were reported for most collected samples, with filaments at various stages of degradation and colonies under possible stress (Fig. 2a-d). In addition to *Trichodesmium*, UCYN-B was the second most abundant diazotroph ranging between 1x10$^2$ and 2x10$^3$ *nifH* copies L$^{-1}$. Other unicellular diazotrophs of the UCYN groups (UCYN-A1 and UCYN-A2) were the least detected diazotrophs (Stenegren et al., 2018). Of the three heterocystous cyanobacterial symbiont lineages (het-1, het-2, het-3), het-1 was the most dominant (1x10$^1$-4x10$^3$ *nifH* copies L$^{-1}$), (Stenegren et al., 2018). Microscopic analyses from LDB demonstrated the co-occurrence of degrading diatom cells, mainly belonging to *Rhizosolenia* (Stenegren et al., 2018) (Fig. 2e-f).

In contrast to LDA and LDB, at LDC, the highest *nifH* copy numbers (up to 6x10$^5$ *nifH* copies L$^{-1}$ at 60 m depth were from the unicellular diazotrophs UCYN-B (Stenegren et al., 2018). *Trichodesmium*

was only detected at 60 m and with very low copy numbers of *nifH* (~$7x10^2$ *nifH* copies $L^{-1}$)
(Stenegren et al., 2018).
Corresponding to the physiological status of the bloom, higher $N_2$ fixation rates (45.0 nmol N $L^{-1}$
$d^{-1}$) were measured in the surface waters (5m) of LDB in comparison with those measured at LDA
and LDC (19.3 nmol N $L^{-1}$ $d^{-1}$ in LDA and below the detection limit at LDC at 5m), (Caffin et al.,
2018).

**3.2. Diazotrophic bloom demise in the LD stations**
Of the 3 long duration stations we examined, LDA and LDB had a higher biomass of diazotrophs
during the 5 days of sampling (section 3.1). Our analyses examining bloom dynamics from the
satellite-derived Chl *a* concentrations indicate a declining trend in chlorophyll-based biomass during
the sampling time period. Yet, both LDA and LDB were still characterized by high (and visible to the
eye at surface) biomass on the first sampling day at each station (day 1) as determined by qPCR and
microscopy (Stenegren et al., 2018). This is different from LDC where biomass was extremely
limited, and no clear evidence was obtained for any specific bloom or bloom demise. We therefore
show results mostly from LDA and LDB and focus specifically on the evidence for PCD and
diazotroph decline in areas with high biomass and surface blooms.
The mortality of phytoplankton at sea can be difficult to discern as it most probably results
from co-occurring processes including physical forces, chemical stressors, grazing, viral lysis, and/or
PCD. Here, we specifically focused on evidence for PCD and whether the influence of zooplankton
grazing on the diazotrophs and especially on *Trichodesmium* at LDA and LDB impacted bloom
dynamics. At LDA and LDB total zooplankton population was generally low. Total zooplankton
population at LDA ranged between 911-1900 individuals $m^{-3}$ and in LDB between 1209-2188
individuals $m^{-3}$ on day 1 and day 5 respectively. *Trichodesmium* is toxic and inedible to most
zooplankton excluding three species of harpacticoid zooplankton- *Macrosettella gracilis, Miracia*
*efferata* and *Oculosetella gracilis* (O'Neil and Roman, 1994). During our sampling days at these
stations, *Macrosettella gracilis* a specific grazer of *Trichodesmium* comprised less than 1 % of the
total zooplankton community with another grazer *Miracia efferata* comprising less than 0.1 % of total
zooplankton community. *Oculosetella gracilis* was not found at these stations. The low number of
harpacticoid zooplankton specifically grazing on *Trichodesmium* found in the LDA and LDB station,
refutes the possibility that grazing caused the massive demise of the bloom. Moreover, the toxicity of
*Trichodesmium* to many grazers (Rodier and Le Borgne, 2008; Kerbrat et al., 2011) could critically
limit the amount of *Trichodesmium*-derived recycled matter within the upper mixed layer.
Viruses have been increasingly invoked as key agents terminating phytoplankton blooms.
Phages may infect *Trichodesmium* (Brown et al., 2013; Hewson et al., 2004; Ohki, 1999) yet they
have not been demonstrated to terminate large surface blooms. Virus-like particles were previously
enumerated from *Trichodesmium* samples during bloom demise, yet the numbers of virus-like
particles did not indicate that a massive, phage-induced lytic event of *Trichodesmium* occurred there
(Spungin et al., 2016). Virus infection may induce PCD by causing an increased production of
reactive oxygen species (Vardi et al., 2012) which stimulates PCD in algal cells (Berman-Frank et al.,
2004; Bidle, 2015; Thamatrakoln et al., 2012). Viral attack can also directly trigger PCD as part of an
antiviral defense system (Bidle, 2015). Virus abundance and activity were not enumerated in this
study, so unfortunately we cannot estimate their specific influence on mortality.
Limited availability of Fe and P induce PCD in *Trichodesmium* (Berman-Frank et al., 2004;
Bar-Zeev et al., 2013). At LDA and LDB, Fe concentrations at the time of sampling were relatively
high (> 0.5 nM), possibly due to island effects (de Verneil et al., 2017). Phosphorus availability, or
lack of phosphorus, can also induce PCD (Berman-Frank et al., 2004; Spungin et al., 2016). $PO_4^{3-}$
concentrations at the surface (0-40m) of LDA and LDB stations were extremely low around 0.05
µmol $L^{-1}$ (de Verneil et al., 2017), possibly consumed by the high biomass and high growth rates of
the bloom causing nutrient stress and bloom mortality. $PO_4^{3-}$ concentrations observed at LDC were
above the quantification limit with average values of 0.2 µmol $L^{-1}$ in the 0-150 m depths (data not
shown). These limited P concentrations may curtail the extent of growth, induce PCD, and pose an
upper limit on biomass accumulation.
Here we compared, for the first time in oceanic populations, two PCD indices, caspase-like
and metacaspase activities, to examine the presence/operation of PCD in the predominant
phytoplankton (and diazotroph) populations along the transect. This was determined by the cleavage
of Z-IETD-AFC and Ac-VRPR-AFC substrates for caspase-like and metacaspase activities
respectively. As we are working with natural communities (and not with monospecific lab cultures),
the activities presented here do not correspond to the purified protein, but to cell free extracts. Thus it
cannot point at the specific cell undergoing PCD or identify the specific organism responsible for the
activity. Here we specifically show the results from LDA and LDB where biomass and activities were
detectable.
Classic caspases are absent in phytoplankton, including in cyanobacteria, and are unique to
metazoans and several viruses (Minina et al., 2017). In diverse phytoplankton the presence of a
caspase domain suffices to demonstrate caspase-like proteolytic activity that occurs upon PCD
induction when the caspase specific substrate Z-IETD-AFC is added (Berman-Frank et al., 2004;
Bidle and Bender, 2008; Bar-Zeev et al., 2013). Cyanobacteria and many diazotrophs contain genes
that are similar to caspases, the metacaspases-cysteine proteases. These proteases share structural
properties with caspases, specifically a histidine-cysteine catalytic dyad in the predicted active site
(Tsiatsiani et al., 2011). While the specific role and function/s of metacaspases genes are unknown,
and cannot be directly linked to gene expression, preliminary investigations have indicated that when
PCD is induced some of these genes are upregulated (Bidle and Bender, 2008; Spungin et al., 2016).
Of the abundant diazotrophic populations at LDA and LDB 12 metacaspases have previously
been identified in *Trichodesmium* spp. (Asplund-Samuelsson et al., 2012; Asplund-Samuelsson, 2015;
Jiang et al., 2010; Spungin et al., 2016). Phylogenetic analysis of a wide diversity of truncated
metacaspase proteins, containing the conserved and characteristic caspase super family (CASc;
cl00042) domain structure, revealed metacaspase genes in both *Richelia intracellularis* (het-1) from
the diatom-diazotroph association and *Crocosphaera watsonii* (a cultivated unicellular
cyanobacterium) which is genetically identical to the UCYN-B *nifH* sequences (Spungin et al.,
unpublished data).
We compared between metacaspase and caspase-like activities for the $> 0.2$ µm fraction
sampled assuming that the greatest activity would be due to the principle organisms contributing to
the biomass – i.e. the diazotrophic cyanobacteria. Caspase-like activity and metacaspase activity were
specifically measured at all LD stations (days 1,3,5) at 5 depths between 0-200 m. Caspase-like
activity at the surface waters (50 m) at LDA, as determined by the cleavage of IETD-AFC substrate,
was between 2.3 to 2.8±0.1 pM hydrolyzed mg protein$^{-1}$ min$^{-1}$ on days 1 and 3 respectively (Fig. 3a).
The highest activity was measured on day 5 at 50 m with 5.1±0.1 pM hydrolyzed mg protein$^{-1}$ min$^{-1}$.
Similar trends were obtained at LDA for metacaspase activity as measured by the cleavage of the
VRPR-AMC substrate, containing an Arg residue at the P1 position, specific for metacaspase
cleavage, (Tsiatsiani et al., 2011). High and similar metacaspase activities were measured on days 1
and 3 (50 m) with 32±4 and-35±0.2 pM hydrolyzed mg protein$^{-1}$ min$^{-1}$ respectively (Fig. 3a). The
highest metacaspase activity was measured on day 5 at 50 m with 59±1 pM hydrolyzed mg protein$^{-1}$
min$^{-1}$ with declining activity at greater depths (Fig. 3b).
Caspase-like activity at LDB, was similar for all sampling days, with the highest activity
recorded from the surface samples (ranging from 3±0.1 to 4.5±0.2 pM hydrolyzed mg protein$^{-1}$ min$^{-1}$
at 7 m depth and then decreasing with depth) (Fig. 3d). At day 3 caspase-like activity at LDB
increased at the surface with 4.5±0.2 pM hydrolyzed mg protein$^{-1}$ min$^{-1}$ and then declined slightly by
day 5 back to 3±0.1 pM hydrolyzed mg protein$^{-1}$ min$^{-1}$. The decrease in activity at the surface between
day 3 and 5 was accompanied by an increase in caspase-like activity measured in the DCM between
day 3 and 5 (Fig. 3d). Caspase-like activity at the DCM at day 3 (35 m) was 1±0.4 pM hydrolyzed mg
protein$^{-1}$ min$^{-1}$ and by day 5 increased to 3±0.1 pM hydrolyzed mg protein$^{-1}$ min$^{-1}$ for samples from 70
m depth. Thus, at LDB, caspase-like activity increased from day 1 to 5 and with depth with Higher
activities were initially recorded at surface and then at depth and were coupled with the decline of the
bloom (Fig. 3d). Similar trends were obtained at LDB for metacaspase activity with 11.1±0.9 pM
hydrolyzed mg protein$^{-1}$ min$^{-1}$ at the surface (7 m) on day 1. A 4-fold increase in activity was
measured at the surface on day 3 with 40.1±5 pM hydrolyzed mg protein$^{-1}$ min$^{-1}$ (Fig. 3e). Similar
high activities were measured also on day 5 (Fig. 3e). However, the increase in activity was also
pronounced at depth of ~ 70 m and not only at the surface. Metacaspase activity at day 5 was the
highest with 40.3±0.5 and 44.6±5 pM hydrolyzed mg protein$^{-1}$ min$^{-1}$ at 7 and 70 m respectively (Fig.
3e). The relatively low metacaspase activity measured on day 1 appears to correspond with the
stressed physiological status of the biomass just prior to increased mortality rates. Metacaspase
activity increased corresponding with the pronounced decline in Chl *a* from day 1 to day 5 (Fig. 1b).
The measured metacaspase activities were typically 10-fold higher than caspase-like activity
rates (Fig. 3). Yet, metacaspase and caspase-like activities are significantly and positively correlated
at LDA and LDB (*r*=0.7, *p*=0.005 and *r*=0.7 *p*=0.001 for LDA and LDB respectively) (Fig. 3c and
3f). Both findings (i.e. higher metacaspase activity and tight correlation between metacaspase and
caspase-like activities) were demonstrated specifically in cultures and natural populations of
*Trichodesmium* undergoing PCD (Spungin et al., unpublished). As our experiments find a significant
positive correlation between both activities, we performed a series of inhibitor experiments to test
whether metacaspases are substrate specific and are not the caspase-like activity we have examined
(Spungin et al., unpublished). In vitro treatment with a metacaspase inhibitor- antipain
dihydrochloride, efficiently inhibited metacaspase activity, confirming the arginine-based specificity
of *Trichodesmium*. Our biochemical activity and inhibitor observations demonstrate that metacaspases
and caspases-like activities are likely distinct and are independently activated under stress and
coupled to PCD in our experiments of both laboratory and field populations. However, caspase-like
activity was somewhat sensitive to the metacaspase inhibitor, antipain, showing a ~30-40% drop in
activity. This hints at some catalytic crossover between these two catalytic activities in
*Trichodesmium* that further should be studied. We do not know what protein is responsible for the
caspase-like specific activities and what drivers regulate it. Yet, the tight correlation between both
activities specifically for *Trichodesmium,* and here at LDA and LDB suggest that both activities occur
in the cell when PCD is induced. To date, we are not aware of any previous studies examining
metacaspase or caspase-like activity (or the existence of PCD) in diatom-diazotroph associations such
as *Rhizosolenia-Richelia*.

### 3.3. TEP dynamics and carbon pools

Transparent exopoloymeric particles (TEP) link between the particulate and dissolved carbon
fractions and act to augment the coagulation of colloidal precursors from the dissolved organic matter
and from biotic debris and to increase vertical carbon flux (Passow, 2002; Verdugo and Santschi,
2010). TEP production also increases upon PCD induction – specifically in large bloom forming
organisms such as *Trichodesmium* (Berman-Frank et al., 2007; Bar-Zeev et al., 2013).
At LDA, TEP concentrations at 50 m depth were highest at day 1 with measured concentrations
of 562±7 µg GX L$^{-1}$ (Table. 1) that appear to correspond with the declining physiological status of the
cells that were sampled at that time (Fig. 2a-d). TEP concentrations during days 3 and 5 decreased to
less than 350 µg GX L$^{-1}$, and it is possible that most of the TEP had been formed and sank prior to our
measurements in the LDA station.

At LDB, TEP concentrations at day 1 and 3 were similar with ~ 400 µg GX L$^{-1}$ at the surface (7

m) while concentrations decreased about 2-fold with depth, averaging at 220±56 and 253±32 µg GX
L$^{-1}$ (35-200 m) for day 1 and 3 respectively (Fig. 4a, Table 2). A significant (> 150 %) increase in
TEP concentrations was observed on day 5 compared to previous days, with TEP values of 597±69 µg
GX L$^{-1}$ at the surface (7m) (Fig. 4b, Table 2). Although TEP concentrations were elevated at surface,
the difference in averaged TEP concentrations observed at the deeper depths (35-200 m) between day
3 (157±28 µg GX L$^{-1}$) and day 5 (253±32 GX L$^{-1}$) indicated that TEP from the surface was either
breaking down or sinking to depth (Fig. 4a, Table 2). The TEP concentrations from this study
correspond with values and trends reported from other marine environments (Engel, 2004; Bar-Zeev
et al., 2009) and specifically with TEP concentrations measured from the New Caledonian lagoon
(Berman-Frank et al., 2016).

TEP are produced by many phytoplankton including cyanobacteria under conditions

uncoupling growth from photosynthesis (i.e. nutrient but not carbon limitation) (Berman-Frank and
Dubinsky, 1999; Passow, 2002; Berman-Frank et al., 2007). Decreasing availability of dissolved
nutrients such as nitrate and phosphate has been significantly correlated with increase in TEP
concentrations in both cultured phytoplankton and natural marine systems (Bar-Zeev et al., 2013;
Brussaard et al., 2005; Engel et al., 2002; Urbani et al., 2005). TEP production in *Trichodesmium* is
enhanced as a function of nutrient stress (Berman-Frank et al., 2007).

In the New Caledonian coral lagoon TEP concentrations were significantly and negatively

correlated with ambient concentrations of dissolved inorganic phosphorus (DIP) (Berman-Frank et al.,
2016). Here, at LDB a significant negative correlation of TEP with DIP was also observed (Fig. 4b,
*p*=0.005), suggesting that lack of phosphorus set a limit to continued biomass increase and stimulated
TEP production in the nutrient-stressed cells. TEP production was also significantly positively
correlated with metacaspase activity at all days (Fig. 4c, *p*=0.03) further indicating that biomass
undergoing PCD produced more TEP. In the diatom *Rhizosolenia setigera* TEP concentrations
increased during the stationary- decline phase (Fukao et al., 2010) and could also affect buoyancy.
Coupling between PCD and elevated production of TEP and aggregation has been previously shown
in *Trichodesmium* cultures (Berman-Frank et al., 2007; Bar-Zeev et al., 2013). Here we cannot
confirm a mechanistic link between nutrient stress, PCD induction, and TEP production, but show
significant correlations between these parameters measured at LDA and LDB with the declining
diazotroph blooms (Fig. 4c) (Spungin et al., 2016).
Furthermore, TEP concentrations at LDB were significantly and positively correlated with TOC,
POC, and DOC (Fig. 4d-f) confirming the integral part of TEP in the cycling of carbon at this station.
Assuming a carbon content of 63 % (w/w), (Engel, 2004) we estimate that TEP contributes to the
organic carbon pool in the order of ~ 80-400 µg C L$^{-1}$ (Table 1 and Table 2) with the percentage of
TEP-C from TOC ranging between 0.08- 42 % and 11-32 % at LDA and LDB respectively (Table 1
and 2, taking into account spatial and temporal differences). Thus, at LDB, surface TEP-C increased
from 22 % at day 3 to 32 % of the TOC content at day 5. Yet, for the same time period a 2-fold
increase of TEP was measured at 200 m (11 % to 21 %). These results reflect the bloom status at
LDB. During bloom development; organic C and N are incorporated to the cells and little biotic TEP
production occurs while stationary growth (as long as photosynthesis continues) stimulates TEP
production (Berman-Frank and Dubinsky, 1999). When mortality exceeds growth, the presence of
large amounts of sticky TEP provide "hot spots" or substrates for bacterial activity and facilitate
aggregation of particles and enhanced sinking rates of aggregates as previously observed for
*Trichodesmium* (Bar-Zeev et al., 2013).

## 3.4. Linking PCD-induced bloom demise to particulate C and N export

Measurements of elevated rates of metacaspase and caspase-like activities and changes in TEP
concentrations are not sufficient to link PCD and vertical export of organic matter as demonstrated for
laboratory cultures of *Trichodesmium* (Bar-Zeev et al., 2013). To see whether PCD-induced mortality
led to enhanced carbon flux at sea we now examined mass flux and specific evidence for diazotrophic
contributions from the drifting sediment traps (150, 325 and 500 m) at LDA and LDB stations.
Mass flux at LDA increased with time, with the maximal mass flux rates obtained from the 150 m
trap (123 dry weight (DW) m$^{-2}$ d$^{-1}$) on day 4. The highest mass flux was 40 and 27 DW m$^{-2}$ d$^{-1}$ from
the deeper sediment traps (325 and 500 m traps respectively). Particulate C (PC) and particulate
nitrogen (PN) showed similar trends as the mass flux. At LDA, PC varied between 3.2-30 mg sample$^{-1}$
$^{1}$ and PN ranged from 0.3-3.2 mg sample$^{-1}$ in the 150 m trap. At LDB, PC varied from 1.6 to 6 mg
sample$^{-1}$ and total PN ranged from 0.2 to 0.8 mg sample$^{-1}$ in the 150 m trap. The total sediment flux in
the traps deployed at LDB ranged between 6.4 mg m$^{-2}$ d$^{-1}$ (150 m, day 4) and 33.5 mg m$^{-2}$ d$^{-1}$ (500 m,
day 2), with an average of 18.9 mg m$^{-2}$ d$^{-1}$. Excluding the deepest trap at 500 m where the high flux
occurred at day 2, in the other traps the highest export flux rate occurred at the last day at the station
(day 5).
Analyses of the community found in the sediment traps, as determined by qPCR from the
accumulated matter on day 5 at the station, confirmed that *Trichodesmium*, UCYN-B and het-1 were
the most abundant diazotrophs in the sediment traps at LDA and LDB stations (Caffin et al., 2018),
significantly correlating with the dominant diazotrophs found at the surface of the ocean (measured on
day 1). *Trichodesmium* and *Rhizosolenia-Richelia* association (het-1) were the major contributors to
diazotroph export at LDA and LDB while UCYN-B and het-1 were the major contributors at LDC
(Caffin et al., 2018). At LDA the deeper traps contained *Trichodesmium* with 2.6 x10$^7$ and 1.4x10$^7$
*nifH* copies L$^{-1}$ at the 325 and 500 m traps respectively. UCYN-B was detected in all traps with the
highest abundance at the 325 m (4.2x10$^6$ *nifH* copies L$^{-1}$) and 500 m traps (2.8x10$^6$ *nifH* copies L$^{-1}$).
Het-1 was found only in the 325 m trap with 2.0x10$^7$ *nifH* copies L$^{-1}$ (Fig. 5a). At LDB,
*Trichodesmium*, UCYN-B and het-1 were detected at the 325 and 500 m traps but not at 150 m.
*Trichodesmium* counts were 9x10$^5$ at the 325 m trap and 5x10$^6$ *nifH* copies L$^{-1}$ for the 500 m trap (Fig.
5b). While evidence for UCYN-B showed 3.6x10$^5$ and 10x10$^5$ *nifH* copies L$^{-1}$ at 325 and the 500 m
traps respectively (Fig. 5b).

In addition to exported *Trichodesmium* and *Rhizosolenia-Richelia* associations, the small

unicellular UCYN-B (< 4 μm) were also found in the sediment traps, including the deeper (500 m)
traps. UCYN-B is often associated with larger phytoplankton such as the diatom *Climacodium*
*frauenfeldianum* (Bench et al., 2013) or in colonial phenotypes (> 10 μm fraction) as has been
observed in the northern tropical Pacific (ALOHA) (Foster et al., 2013). Sedimenting UCYN-B were
detected during the VAHINE mesocosm experiment in the New Caledonian lagoon in shallow (15m)
sediment traps) (Bonnet et al., 2015) and were also highly abundant in a floating sediment trap
deployed at 75 m for 24 h in the North Pacific Subtropical Gyre (Sohm et al., 2011). Thus our data
substantiates earlier conclusions that UCYN, which form large aggregates (increasing actual size and
sinking velocities), can efficiently contribute to export in oligotrophic systems (Bonnet et al., 2015).
Increase in aggregate size could also occur with depth, possibly due to the high concentrations of TEP
produced at the surface layer that provide a nutrient source and enhance aggregation as they sink
down the water column (Berman-Frank et al., 2016).

The sinking rates of aggregates in the water column, depend on factors such as fluid viscosity,

particle source material, morphology, density, and variable particle characteristics. Sinking velocities
of diatoms embedded in aggregates are generally fast (50-200 m d$^{-1}$) (Asper, 1987; Alldredge, 1998)
compared with those of individually sinking cells (1$^{-10}$ m d$^{-1}$) (Culver and Smith, 1989) allowing
aggregated particles to sink out of the photic zone to depth. Assuming a sinking rate of
*Trichodesmium*-based aggregates of 150-200 m d$^{-1}$ (Bar-Zeev et al., 2013), we would need to shift the
time frame by 1 day to see whether PCD measured from the surface waters is coupled with changes in
organic matter reflected in the 150 m sediment traps. Thus, at LDA, examining metacaspase activities
from the surface with mass flux and particulate matter obtained 24 h later yielded a significant
positive correlation between these two parameters (Fig. 5c).

LDA had the highest export flux and particulate matter found in its traps relative to LDB and

LDC. Diazotrophs contributed ~ 36 % to PC export in the 325 m trap at LDA, with *Trichodesmium*
comprising the bulk of diazotrophs (Caffin et al., 2018). In contrast, at LDB, we found lower flux
rates and lower organic material in the traps. *Trichodesmium* contributed the bulk of diazotroph
biomass at the 150 m trap. We believe that at LDB the decline phase began only halfway through our
sampling and thus the resulting export efficiency we obtained for the 5 days at station was relatively
low compared to the total amount of surface biomass. Moreover, considering export rates, and the
experimental time frame, most of the diazotrophic population may have been directly exported to the
traps only after we left the station (i.e. time frame > 5 days). This situation is different from the bloom
at LDA, where enhanced mortality, biomass deterioration, and bloom crash were initiated 1-2 weeks
before our arrival and sampling at the station. Thus, at LDA, elevated mass flux and higher
concentrations of organic matter were obtained from all three depths of the deployed traps.
In the field, especially in the surface layers of the oligotrophic oceanic regions, dead cells are
rarely seen at later stages (Berges and Choi 2014; Segovia et al., 2018). This is due to the fact that
dying and dead cells are utilized quickly and recycled within the food web and upper surface layer.
However, under bloom conditions, when biomass is high, the fate of the extensive biomass is more
complicated (Bonnet et al., 2015). PCD induced cell death, combined with buoyancy loss, can lead to
rapid sinking to depth of the biomass at a speed that would prevent large feeding events on this
biomass. We previously measured POC export in our laboratory under controlled conditions (Bar-
Zeev et al., 2013). Here, using sediment traps we measured POC fluxes as well as specific indices
(*NifH* reads) of *Trichodesmium* and other diazotrophs which were measured for several days at the
surface where high biomass accumulations were found. This indicates that under bloom conditions
when biomass is high some of the cell pellets do sink down out of the food web.

## 4. Conclusion and implications

Our specific objective in this study was to examine whether diazotroph mortality mediated by
PCD can lead to higher fluxes of organic matter sinking to depth. The OUTPACE cruise provided this
opportunity in two out of three long-duration (5 day) stations where large surface blooms of
diazotrophs principally comprised of *Trichodesmium*, UCYN-B and diatom-diazotroph associations
*Rhizosolenia-Richelia* were encountered. We demonstrate (to our knowledge for the first time)
metabolically active metacaspases in oceanic populations of *Richelia* and *Trichodesmium*. Moreover,
metacaspase activities were significantly correlated to caspase-like activities at both LDA and LDB
stations. Both caspase and metacaspase-proteins families are independent yet characteristic of PCD
induced mortality. Evidence from drifting sediment traps, deployed for 5 days at the two stations,
showed high TEP concentrations formed at surface and shifting to depth, increasing numbers of
diazotrophs in sediment traps from 150, 350, 500 m depths), and a time-shifted correlation between
metacaspase activity (signifying PCD) and vertical fluxes of PC and PN).
Yet, our results also delineate the natural variability of biological oceanic populations. The two
stations, LDA and LDB were characterized by biomass at physiologically different stages. The
biomass from LDA displayed more pronounced mortality that had begun prior to our arrival at station.
In contrast, satellite data indicated that at LDB, the surface *Trichodesmium* bloom was sustained for at
least a month prior to the ship's arrival and remained high for the first 3 days of our sampling before
declining by 40 % at day 5. As sediment trap material was examined during a short time frame, of
only 5 days at each LD station, we assume that a proportion of the sinking diazotrophs and organic
matter were not yet collected in the traps and had either sunk before trap deployment or would sink
after we left the stations. Thus, these different historical conditions, which influence physiological
status at each location, also impacted the specific results we obtained and emphasized a-priori the
importance of comprehensive spatial and temporal sampling that would facilitate a more holistic
understanding of the dynamics and consequences of bloom formation and fate in the oceans.

**Author contributions**

IBF, DS, and SB conceived and designed the investigation linking PCD to vertical flux within the
OUTPACE project. NB, MS, AC, MPP, NL CD and RAF participated, collected and performed
analyses of samples, DS analysed samples and data. DS and IBF wrote the manuscript with
contributions from all co-authors.

**Acknowledgments**

This research is a contribution of the OUTPACE (Oligotrophy from Ultra-oligoTrophy PACific
Experiment) project (https://outpace.mio.univ-amu.fr/) funded by the Agence Nationale de la
Recherche (grant ANR-14-CE01-0007-01), the LEFE-CyBER program (CNRS-INSU), the Institut de
Recherche pour le Développement (IRD), the GOPS program (IRD) and the CNES (BC T23, ZBC
4500048836). The OUTPACE cruise (http://dx.doi.org/10.17600/15000900) was managed by the
MIO (OSU Institut Pythéas, AMU) from Marseilles (France). The authors thank the crew of the R/V
L'Atalante for outstanding shipboard operations. G. Rougier and M. Picheral are warmly thanked for
their efficient help in CTD rosette management and data processing, as well as C. Schmechtig for the
LEFE-CyBER database management. Aurelia Lozingot is acknowledged for the administrative work.
All data and metadata are available at the following web address: http://www.obs-
vlfr.fr/proof/php/outpace/outpace.php. We thank Olivier Grosso (MIO) and Sandra Hélias (MIO) for
the phosphate data and François Catlotti (MIO) for the zooplankton data. The ocean color satellite
products were provided by CLS in the framework of the CNES-OUTPACE project (PI A.M. Doglioli)
and the video is courtesy of A. de Verneil. RAF acknowledges Stina Höglund and the Image Facility
of Stockholm University and the Wenner-Gren Institute for access and assistance in confocal
microscopy. The participation of NB, DS, and IBF in the OUTPACE experiment was supported
through a collaborative grant to IBF and SB from Israel Ministry of Science and Technology Israel
and the High Council for Science and Technology (HCST)-France 2012/3-9246, and United States-
Israel Binational Science Foundation (BSF) grant No. 2008048 to IBF. RAF was funded by the Knut
and Alice Wallenberg Stiftelse, and acknowledges the helpful assistance of Dr. Lotta Berntzon. This
work is in partial fulfillment of the requirements for a PhD thesis for D. Spungin at Bar-Ilan
University.

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

     **Figure legends**

**Figure 1-** Temporal dynamics of surface chlorophyll *a* (Chl *a*) concentrations in the long duration (LD) stations **(a)** LDA **(b)** LDB and **(c)** LDC station. Chl *a* was measured over 5 days at each station (marked in gray). Satellite data of daily surface Chl *a* (mg m$^{-3}$) around the LD stations of OUTPACE was used to predict changes in photosynthetic biomass before and after our measurements at the station (marked as dashed lines). Satellite data movies are added as supplementary data (Supplementary videos S1, S2, S3). Chl *a* profiles in **(d)** LDA **(e)** LDB and **(f)** LDC. Measurements of Chl *a* were taken on days 1 (black dot), 3 (white triangle) and 5 (grey square) at the LDB station at 5 depths between surface and 200 m depths.

**Figure 2- (a-d)** Microscopic images of *Trichodesmium* from LDA and LDB. Observations of poor cell integrity were reported for collected samples, with filaments at various stages of degradation and colony under possible stress. **(e)** Confocal and **(f)** processed IMARIS images of *Rhizosolenia-Richelia* symbioses (het-1) at 6m (75 % surface incidence). Green fluorescence indicates the chloroplast of the diatoms, and red fluorescence are the *Richelia* filaments; Microscopic observations indicate that near the surface *Rhizosolenia* populations were already showing signs of decay since the silicified cell-wall frustules were broken and free filaments of *Richelia* were observed. Images by Andrea Caputo.

**Figure 3- PCD indices from LDA and LDB (a)** Caspase-like activity from LDA (pM hydrolyzed mg protein$^{-1}$ min$^{-1}$) assessed by cleavage of the canonical fluorogenic substrate, z-IETD-AFC. **(b)** Metacaspase activity from LDA (pM hydrolyzed mg protein$^{-1}$ min$^{-1}$) assessed by cleavage of the canonical fluorogenic substrate, Ac-VRPR-AMC. **(c)** Relationship between caspase-like activity and metacaspase activity from LDA ($r$=0.7, n=15, $p$=0.005). **(d)** Caspase-like activity rats in LDB station (pM hydrolyzed mg protein$^{-1}$ min$^{-1}$). **(e)** Metacaspase activity in LDB station (pmol hydrolyzed mg protein$^{-1}$ min$^{-1}$). **(f)** Relationship between caspase-like activity and metacaspase activity in LDB station ($r$=0.7, n=15, $p$=0.001). Caspase-like and metacaspase activities at LDA and LDB stations were measured on days: 1 (black dot), 3 (white triangle) and 5 (grey square) between surface and 200 m. Error bars represent ± 1 standard deviation (n=3).

**Figure 4- (a)** Depth profiles of TEP concentrations (µg GX L$^{-1}$) at LDB station. Measurements were taken on days 1, 3 and 5 at the station at surface-200 m depths. **(b)** The relationships between the concentration of transparent exopolymeric particles (TEP), (µg GX L$^{-1}$) and dissolved inorganic phosphorus DIP (µmol L$^{-1}$) for days 1, 3 and 5 at the LDB station ($r$=-0.7, n=15, $p$=0.005). Relationships between the concentration of transparent exopolymeric particles (TEP), (µg GX L$^{-1}$) and **(c)** metacaspase activity (pmol hydrolyzed mg protein$^{-1}$ min$^{-1}$) for days 1, 3 and 5 at the LDB assessed by cleavage of the canonical fluorogenic substrate, Ac-VRPR-AMC ($r$ =0.6 n=15, $p$=0.03);

**(d)** and with dissolved organic carbon (DOC), (µM) for days 1, 3 and 5 at the LDB station ($r$=0.7,
n=15, $p$=0.004) **(e)** and with particulate organic carbon (POC) (µM) for days 1, 3 and 5 at the LDB
station ($r$=0.8, n=5, $p$=0.1 for day 1 and $r$=0.9, n=8 $p$=0.002 for day 3and 5) **(f)** and with total organic
carbon (TOC) (µM) for days 1, 3 and 5 at the LDB station ($r$=0.7, n=15, $p$=0.001). Measurements
were taken on days 1 (black dot), 3 (white triangle) and 5 (grey square) at LDB at 5 depths between
surface and 200 m depths. Error bars for TEP represent ± 1 standard deviation (n=3)**.**

**Figure 5- (a)** Diazotrophic abundance (*nifH* copies L$^{-1}$) of *Trichodesmium* (dark grey bars); UCYN-B
(white bars); and het-1 (light grey bars) recovered in sediment traps at the LDA station. **(b)**
Diazotrophic abundance (*nifH* copies L$^{-1}$) of *Trichodesmium* (dark grey bars); UCYN-B (white bars);
and het-1 (light grey bars) recovered in sediment traps at the LDB station. Abundance was measured
from the accumulated material on day 5 at each station. Sediment traps were deployed at the LD
station at 150 m, 325 m, and 500 m. Error bars represent ± 1 standard deviation (n=3). **(c)**
Relationship between metacaspase activity (pmol hydrolyzed mg protein$^{-1}$ min$^{-1}$) measured at the
surface waters of LDA station assessed by cleavage of the canonical fluorogenic substrate, Ac-VRPR-
AMC and mass flux rates (mg m$^2$ h$^{-1}$) (grey circle), particulate carbon (PC, mg sample$^{-1}$) (green
triangle) and particulate nitrogen (PN, mg sample$^{-1}$) (blue square) measured in the sediment trap
deployed at 150 m. A 1-day shift between metacaspase activities at the surface showed a significant
positive correlation with mass flux and particulate matter obtained in the sediment trap at LDA station
at 150 m.














**Table 1**- Temporal changes in the relative composition (w/w) and distribution of TEP, TEP-C and
organic carbon and nitrogen fractions within the water column during days 1,3 and 5 in the LDA
station at different depth ranging between surface (10 m) to 200 m.

| Day at LDA station | Depth (m) | TEP (µg GX L⁻¹) | TEP-C | %TEP-C | POC (µM) | TOC (µM) | POC/PON |
|---|---|---|---|---|---|---|---|
| 1 | 200 | 296±135 | 186.5 | 27.2 | 3.04 | 57.2 | 5 |
| | 150 | ND | ND | ND | 3.18 | 61.1 | 13 |
| | 70 | 87±17 | 54.8 | 6.7 | 2.93 | 68.7 | 11 |
| | 50 | 562±7 | 354.3 | 41.9 | 2.47 | 70.5 | 13 |
| | 10 | 241±40 | 152.3 | 14.5 | 9.21 | 87.4 | 8 |
| 3 | 200 | 191±13 | 120.9 | 18.6 | 1.29 | 54.2 | 27 |
| | 150 | 144±54 | 91.2 | 12.9 | 2.22 | 59.0 | 22 |
| | 80 | 263 | 166.1 | 20.5 | 4.62 | 67.5 | 15 |
| | 10 | 126±2 | 79.6 | 8.3 | 3.60 | 79.7 | 12 |
| 5 | 200 | 200 | 126 | 21.3 | 2.84 | 54.2 | 236 |
| | 150 | 220 | 138.6 | 18.0 | 2.72 | 58.2 | 7 |
| | 80 | 146 | 92.2 | 12.1 | 4.91 | 63.3 | 8 |
| | 50 | 348±60 | 219.5 | 26.8 | 3.33 | 68.3 | 6 |
| | 10 | ND | ND | ND | 5.80 | 83.7 | 7 |


**Table 2-** Temporal changes in the relative composition (w/w) and distribution of TEP, TEP-C and
organic carbon and nitrogen fractions within the water column during days 1,3 and 5 in the LDB
station at different depth ranging between surface (7 m) to 200 m.

| Day at LDB station | Depth (m) | TEP (μg GX L$^{-1}$) | TEP-C | %TEP-C | POC (μM) | TOC (μM) | POC/PON |
|---|---|---|---|---|---|---|---|
| 1 | 7 | 408±36 | 257.1 | 23.4 | 8.95 | 91.5 | 6.0 |
| | 35 | 279±86 | 175.9 | 17.0 | 5.86 | 86.0 | 9.1 |
| | 100 | 214±67 | 134.7 | 16.8 | ND | 66.7 | ND |
| | 150 | 145±34 | 91.5 | 12.3 | 3.79 | 61.9 | 11.2 |
| | 200 | 244±113 | 153.7 | 20.3 | 7.61 | 63.2 | 9.8 |
| 3 | 7 | 402±12 | 253.1 | 22.5 | 8.88 | 93.9 | 6.9 |
| | 35 | 193±48 | 121.8 | 12.6 | 3.07 | 80.3 | 8.2 |
| | 100 | 163±33 | 102.4 | 12.6 | ND | 67.8 | ND |
| | 150 | 145±34 | 91.6 | 12.0 | 1.91 | 63.8 | 7.4 |
| | 200 | 127±79 | 80.2 | 11.3 | 1.71 | 59.3 | 5.7 |
| 5 | 7 | 565±87 | 355.8 | 32.5 | 5.32 | 91.3 | 5.9 |
| | 70 | 294±53 | 185.2 | 20.1 | 2.21 | 76.7 | 6.1 |
| | 100 | 264±160 | 166.2 | 19.6 | 2.25 | 70.6 | 8.0 |
| | 150 | 224±51 | 140.8 | 15.9 | 1.53 | 73.9 | 5.1 |
| | 200 | 231±45 | 145.8 | 21.1 | 1.11 | 57.6 | 5.5 |


Abbreviations: TEP, transparent exopolymeric particle; TEP-C, TEP carbon; POC, particulate organic
C; TOC, total organic C; ND- no data.


**Figures**
**Figure 1**

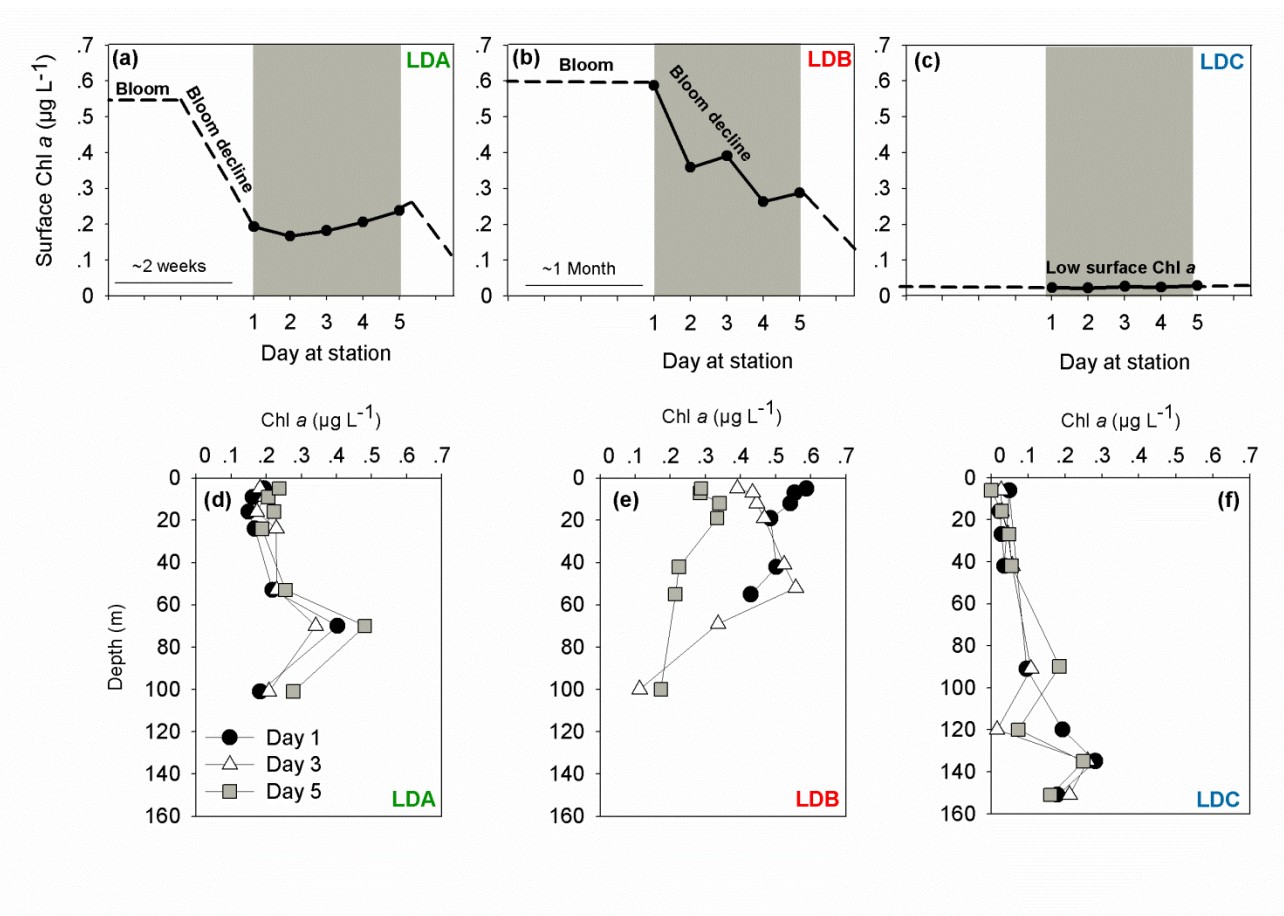








**Figure 2**

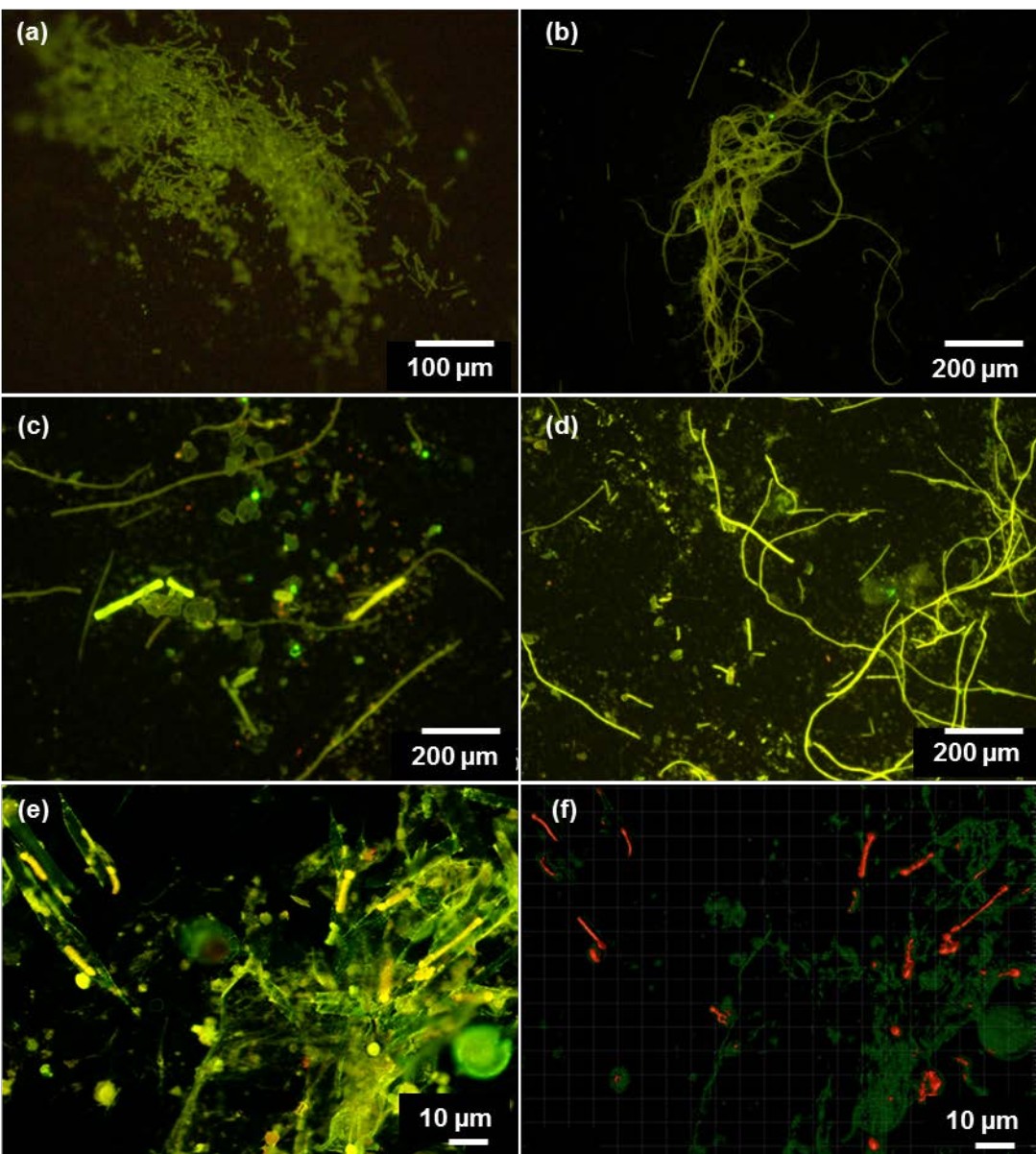






**Figure 3**

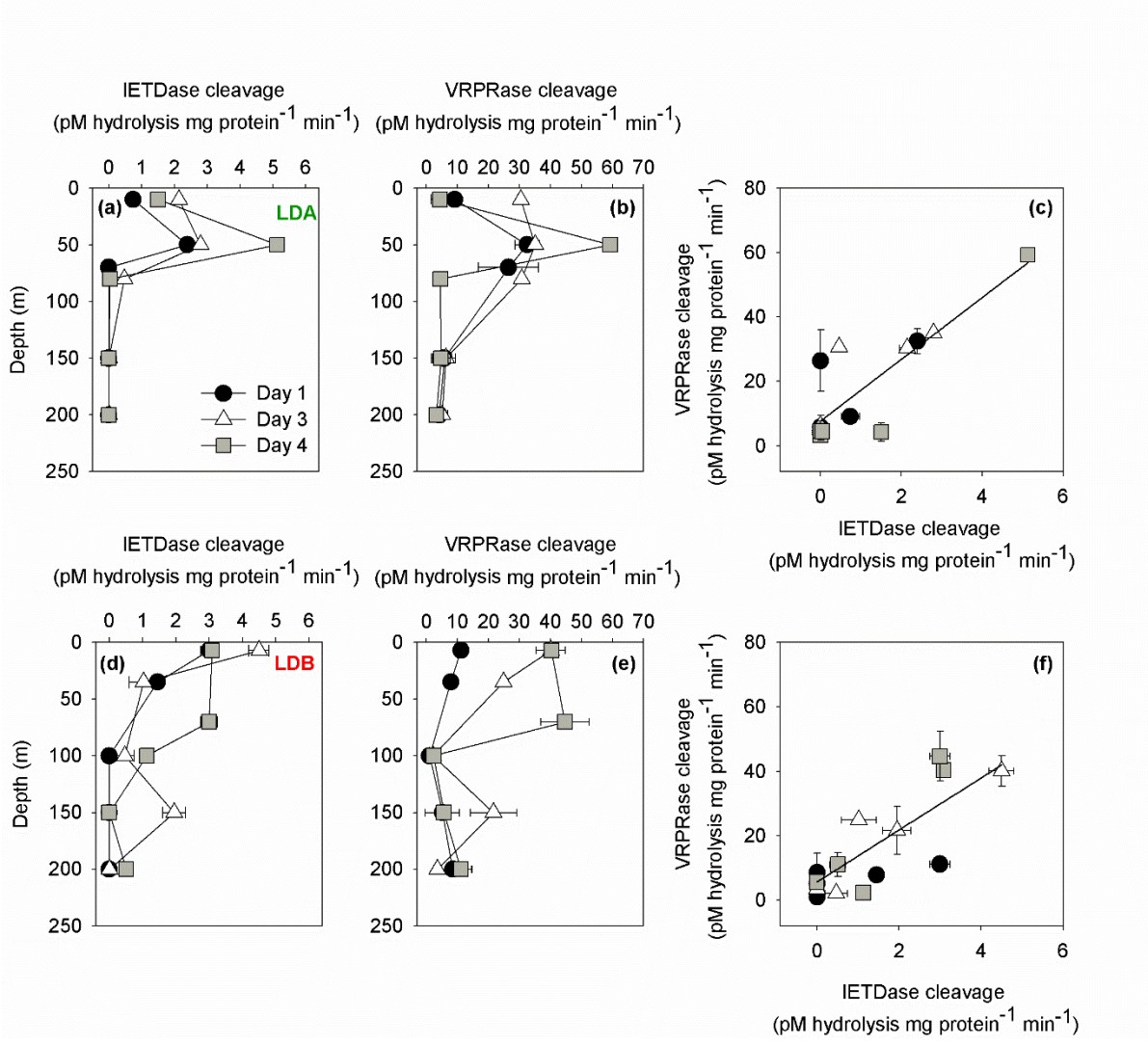

**Figure 4**

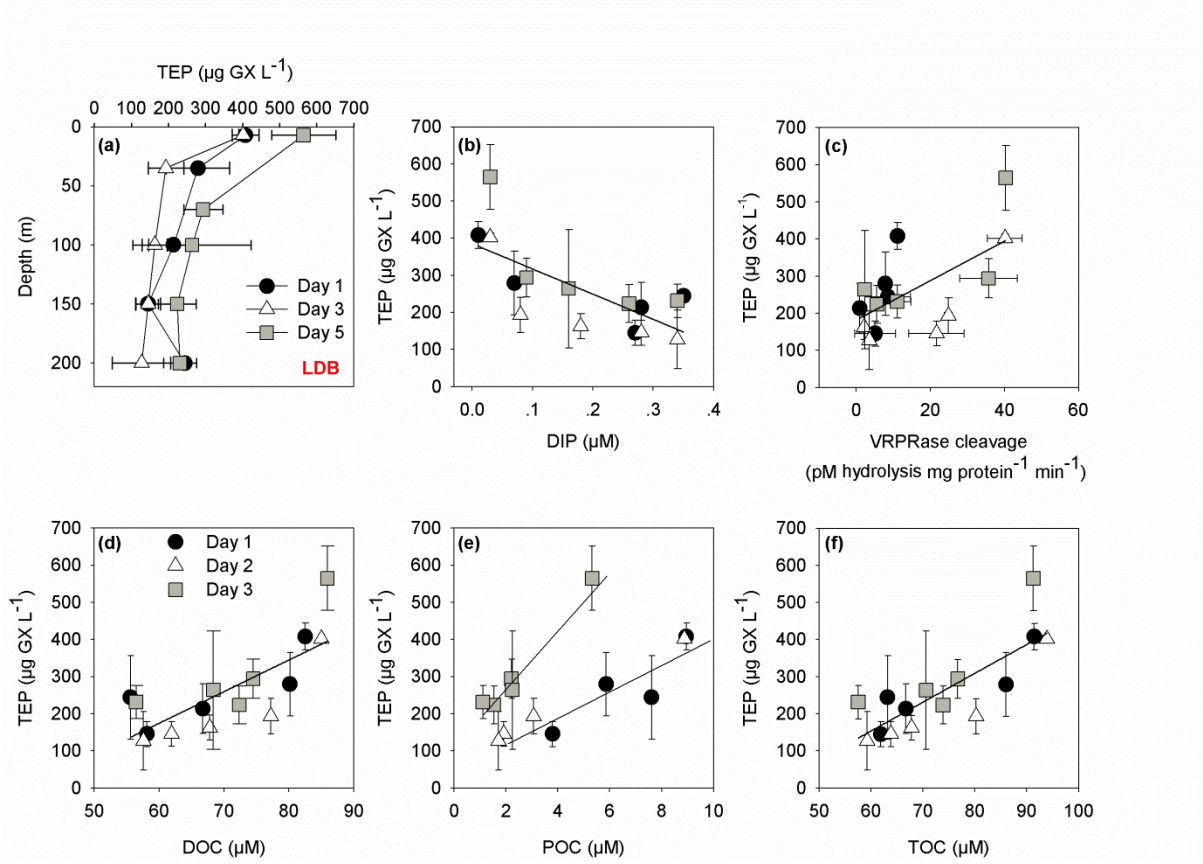







**Figure 5**

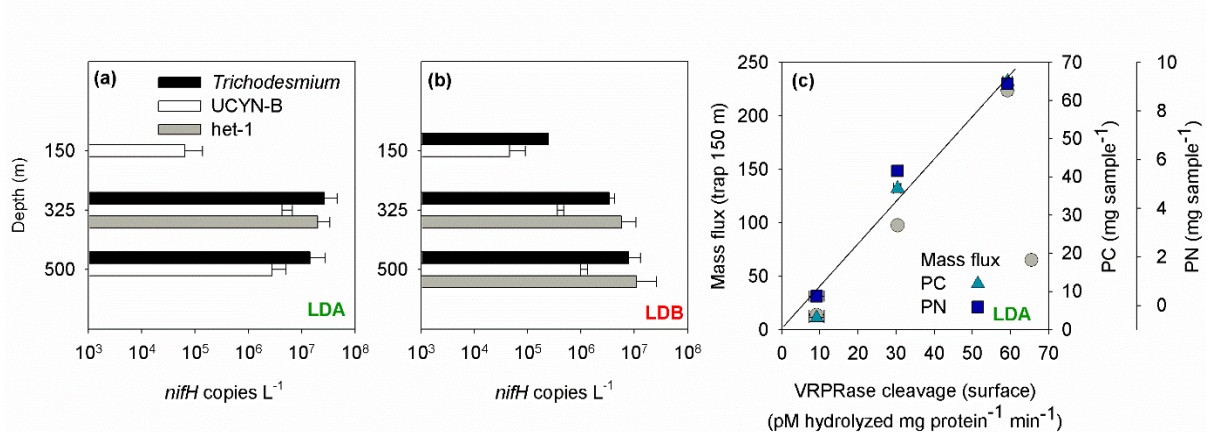

