# Peer review of "Programmed cell death in diazotrophs and the fate of organic matter in the western tropical South Pacific Ocean during the OUTPACE cruise"

_Biogeosciences, 2018_

## Referee Comment (RC1) · Anonymous Referee #1 · 14 Feb 2018

This paper addresses an important question in biological oceanography: the death of phytoplankton. Despite it has been challenged in many instances, current belief is still that phytoplankton are immortal. This study contributes to the growing evidence that other loss mechanisms occur apart from grazing and sinking. The link to falling carbon is indeed very relevant for the global C cycle in the oceans. The manuscript presents excellent science, appropriate methods and conclusions well supported by the data. The presentation is very good, concise and well-structured In my opinion the manuscript can be accepted as it is

---

## Referee Comment (RC2) · Anonymous Referee #2 · 15 Mar 2018

INTERACTIVE COMMENT on "Programmed cell death in diazotrophs and the fate of organic matter in the Western Tropical South Pacific Ocean during the OUTPACE cruise" By Spungin et al.

GENERAL COMMENTS: The manuscript by Spungin et al. reports on the induction of cell death by nutrient limitation during blooms of the diazotroph Trichodesmium (and also diazotrophs associated to diatoms) during a cruise in the South Pacific Ocean. The study analyses whether cell death is a relevant mechanism driving Trichodesmium mortality , and whether this contributes to vertical export of organic matter.

The aim of this work is to provide evidence of correlation between bloom termination, cell death and vertical export, with the mediation of specific proteases such as caspase-like and metacaspases. Trichodesmium is responsible for roughly half of the nitrogen fixed in the ocean. The study hereby presented focuses on a relevant topic for marine biologists which helps elucidate the impact of cell death of a globally relevant species on the ecosystem, warranting important consequences for the C and N cycles. The paper clearly disserves publication since most of the conclussion are strong.

However, I have several concerns that (in my opinion) need to be addressed by the authors before it can be published.

SPECIFIC COMMENTS:

A) I have two major concerns. The first, relates to gene expression and activity of metacaspases and caspase like-proteins and their role in a death cascade (initiation and execution of PCD).

1. The mechanisms by which cell death (CD) (programmed or not programmed) occurs, considering that cell death in phytoplankton leads to the complete demise of the organism/colonies, are always intriguing and there still are many unanswered questions. Among them, which is the proteolytic machinery involved and how it works. Metacaspases, belong to the CD clan of cysteine proteases, were thought to perform similar functions than caspases. It has been shown by multiple labs working with several organisms from yeast, plants and protists that metacaspases are quite distinct in terms of target site specificity from caspases . They target substrate sites are either arginine (R) or lysine (V) at the P1 position. The authors consider this approach right and use a substrate typically hydrolysed by MCs. I was wondering why this specific (VRPR) substrate was used and no other? and, why in the concentration described, 50mM?

Did not the authors test for the optimal substrate concentrations for Trichodesmium before the analyses? The reference they give is based on Arabidopsis thaliana assays and that certainly is very different to cyanobacteria. Clarification is needed.

2.-Same applies with the caspase like substrate IETD, but in this case, I assume that this has been previously tested according to Berman-Frank and Baar-Zev former studies. Caspase -like activities have been reported in vascular plants, phytoplankton, yeast and protozoa. However, their nature is controversial . Up to date, is still not clear, who is the responsible for the observed caspase-like activity in phytoplankton. In vascular plants some authors have pointed to the serine protease family proteins to perform this hydrolysis (see Bonneau et al., 2008) and/or the vacuolar processing enzyme (Hara-Nishimura and Hatsugai, 2011). It has also been reported that some caspase-like activities are attributable to the plant subtilisin-like proteases-saspases and phytaspases (see Vartapetian et al 2011). Hence, clarify this in the text please.

To me the question is:

Since we are measuring these enzymatic activities in phytoplankton's cell free extracts and not in purified proteins result of gene overexpression, we shall be very careful when ascribing the activity to a species. What I mean is : in a cell free extract there are many proteins potentially users of the mentioned substrates. For this reason, I find the use the term "caspase" is not correct, but instead use the term "caspase-like" throughout the whole MS. It is appropriate that the activity must be referred to as "IETDase, etc Therefore, substitute "caspase activity" by "Caspase-like" ( or CL). The same applies to metacaspases, and so VRPRase must be used. Otherwise it can lead to confusion. By the way, revise the nomenclature of the substrate: "Av-VRPR", what group linked to the peptide is Av? Could possible be that Av is in reality Ac?

3.-Along the same thought, the gene expression measurement is very important, but I must say, that does not mean that the enzymatic activity you are measuring corresponds to the expressed gene , if , as said before, that specific activity has not been measured in a purified protein. Hence, caution is needed on this respect when interpreting your data.

4.-Aditionally, I think we all must accept that we do not really know if there are initiator or executor CLs or MCs in phytoplankton. Two types of metacaspases (types I and II) are defined based on the presence of a prodomain analogous to the classification of caspases into initiator or executioner caspases. The molecular role of a prodomain in initiator caspases is the recruitment of caspases to multicomponent signalling complexes for caspase activation. However, phytoplankton metacaspases often lack prodomains (Choi and Berges 2013 ). As I see it, to use this homology can lead into mistake, so I would not describe the enzymes involved as executors of the cell, or initiators of the cascade (although for vascular plants is widely used, it is different, they know exactly which protease which is, and what they do).

5.-Last but not least, just would like to know your opinion on this actual heated-debate: Do you think that at the time being caspase-like proteins, in phytoplankton, could hydrolyse R or V ?

B) The second major concern relates to the fact of bloom/ cell dismissal in the water column.

1.-When working in the field, dead cells are rarely seen at later stages (Berges and Choi 2014) or not seen (Segovia et al., 2018), only because they have been cleared away from the system. Any source of energy that cellular debris may provide to the neighbourhood will be immediately used by other species within the food web. So, it is very unlikely to see cellular rests consequence of CD on the water column. Yet, POC downward flux is the way to have some estimates. In my opinion and experience, this can be applied to cultures in the lab under controlled conditions, but I find it truly complicated in natural communities / ecosystem level. Please, clarify how this fits within your sampling/sample analyses time framework . Has that to do with the blooming condition excluding other components of the trophic web of the niche?

2.- Nothing is said about viruses affecting C losses, which is important for C cycling and definitively affect C export. Viruses were not measured the text says. But in my opinion, this shall at least be discussed and do not directly exclude this possibility as a

possible cause for bloom demise. Is there any long-term study done on Trichodesmium blooms termination affected by viruses that at least allows you to compare with other situations?

BIBLIOGRAPHY: Bonneau L, Ge Y, Drury GE, Gallois P. 2008. What happened to plant caspases? Journal of Experimental Botany 59, 491-499).

Vartapetian AB, Tuzhikov AI, Chichkova NV, Taliansky M, Wolpert TJ (2011) A plant alternative to animal caspases: subtilisin-like proteases. Cell Death Differ 18:1289-1297

Choi CJ, Berges JA. New types of metacaspases in phytoplankton reveal diverse origins of cell death proteases . Cell Death and Disease (2013) 4, e490; doi:10.1038/cddis.2013.21

Berges JA Choi CJ. Cell death in algae: physiological processes and relationships with stress (2014).Perspectives in Phycology .1:103-112

Segovia, M., Lorenzo, M.R., Iñiguez, C., García-Gómez C., (2018). Physiological stress response associated with elevated CO2 and dissolved iron in a phytoplankton community dominated by the coccolithophore Emiliania huxleyi. Mar. Ecol. Prog. Ser. 586-73-89

---

## Author Comment (AC1) · 4 Apr 2018

[revised manuscript text omitted]

---

## Author Comment (AC2) · 4 Apr 2018

The authors would like to thank referee #2 for helpful and insightful comments for the improvement of our manuscript. Attached as a supplementary file (zip) is the response to referee #2 as well as a revised version of the manuscript.

Please also note the supplement to this comment:
https://www.biogeosciences-discuss.net/bg-2018-3/bg-2018-3-AC2-supplement.zip

---

## Author Response (AR1)

**1 Response to reviewer #2**

We would like to thank referee #2 for his insightful comments for clarification and improvement of the manuscript. We have addressed all comments and questions and have revised the manuscript accordingly. Our answers follow the comments in brown.

A marked-up manuscript version with the relevant changes made is also included.

GENERAL COMMENTS: The manuscript by Spungin et al. Reports on the induction of cell 9 death by nutrient limitation during blooms of the diazotroph *Trichodesmium* (and also 10 diazotrophs associated to diatoms) during a cruise in the South Pacific Ocean. The study 11 analyses whether cell death is a relevant mechanism driving *Trichodesmium* mortality, and 12 whether this contributes to vertical export of organic matter. The aim of this work is to 13 provide evidence of correlation between bloom terminations, cell death and vertical 14 export, with the mediation of specific proteases such as caspase-like and 15 metacaspases. *Trichodesmium* is responsible for roughly half of the nitrogen fixed in 16 the ocean. The study hereby presented focuses on a relevant topic for marine biologists 17 which helps elucidate the impact of cell death of a globally relevant species on the 18 ecosystem, warranting important consequences for the C and N cycles. The paper 19 clearly disserves publication since most of the conclusion are strong.

However, I have several concerns that (in my opinion) need to be addressed by the authors
before it can be published.

**23 SPECIFIC COMMENTS:**

I have two major concerns. The first, relates to gene expression and activity of metacaspases and caspase like-proteins and their role in a death cascade (initiation and
 execution of PCD).

As a general comment important to many of the comments below, we would like to note that we have a submitted manuscript in review currently in Environmental Microbiology examining in detail the expression and activity of metacaspases and caspase-like proteins and their involvement in PCD in *Trichodesmium* (This manuscript can be sent if requested).

1. The mechanisms by which cell death (CD) (programmed or not programmed) occurs, 33 considering that cell death in phytoplankton leads to the complete demise of the 34 organism/colonies, are always intriguing and there still are many unanswered questions. 35 Among them, which is the proteolytic machinery involved and how it works. 36 Metacaspases, belong to the CD clan of cysteine proteases, were thought to perform 37 similar functions than caspases. It has been shown by multiple labs working with several 38 organisms from yeast, plants and protists that metacaspases are quite distinct in terms of 39 target site specificity from caspases. They target substrate sites are either arginine (R) or 40 lysine (V) at the P1 position. The authors consider this approach right and use a substrate 41 typically hydrolyzed by MCs. I was wondering why this specific (VRPR) substrate was used 42 and no other? and, why in the concentration described, 50mM? Did not the authors test 43 for the optimal substrate concentrations for *Trichodesmium* before the analyses? The 44 reference they give is based on Arabidopsis thaliana assays and that certainly is very 45 different to cyanobacteria. Clarification is needed. Same applies with the caspase like 46 substrate IETD, but in this case, I assume that this has been previously tested according to 47 Berman-Frank and Bar-Zev former studies.

We chose this specific substrate as recommended in Tsiatsiani et al., 2011 as a fluorogenic 49 substrate with a Arg residues at the P1 position to specifically detect metacaspase activities in 50 cellular extracts. This substrate was experimentally tested with our Trichodesmium cultures and 51 was found to suit our purpose. All results and method discussion are currently in review in a paper 52 we have recently submitted to Environmental Microbiology (Spungin et al., in review EM). We 53 used this specific concentration (50 mM) as an equivalent concentration to the IETD used for the 54 determination of caspase-like activity which has been shown to be the optimal concentrations on 55 cell extracts. This specific substrate was also checked and calibrated pre-experiments (Spungin et 56 al., in review EM). After calibration, we first applied this method in laboratory experiments under 57 controlled conditions and then in natural samples collected from a bloom in the New Caledonian lagoon. The use of this method during the OUTPACE cruise is after calibration and work on other 58 59 experiments. To our knowledge we are the first to use specific metacaspase substrates to test 60 direct metacaspase activity in phytoplankton.

**2.** Caspase -like activities have been reported in vascular plants, phytoplankton, yeast and 63 protozoa. However, their nature is controversial. Up to date, is still not clear, who is the 64 responsible for the observed caspase-like activity in phytoplankton. In vascular plants 65 some authors have pointed to the serine protease family proteins to perform this hydrolysis (see Bonneau et al., 2008) and/or the vacuolar processing enzyme (Hara-66 67 Nishimura and Hatsugai, 2011). It has also been reported that some caspase-like activities 68 are attributable to the plant subtilisin-like proteases-saspases and phytaspases (see 69 Vartapetian et al 2011). Hence, clarify this in the text please. **To me the question is:** Since 70 we are measuring these enzymatic activities in phytoplankton's cell free extracts and not 71 in purified proteins result of gene over expression, we shall be very careful when ascribing 72 the activity to a species. What I mean is: in a cell free extract there are many proteins 73 potentially users of the mentioned substrates. For this reason, I find the use the term 74 "caspase" is not correct, but instead use the term "caspase-like" throughout the whole 75 MS. It is appropriate that the activity must be referred to as "IETDase, etc Therefore, 76 substitute "caspase activity" by "Caspase-like" (or CL). The same applies to 77 metacaspases, and so VRPRase must be used. Otherwise it can lead to confusion.

We completely agree with the reviewer that as *Trichodesmium* does not have true caspases, the
correct form throughout should be "caspase-like" and have corrected this throughout the text to
caspase-like activity. In the figures and legend, we have changed nomenclature to reflect the
specific substrate: i.e. IETDase or VRPRase cleavage (Figures 3, 4, and 5 in the manuscript).

**3.** By the way, revise the nomenclature of the substrate: "Av-VRPR", what group linked
to the peptide is Av? Could possible be that Av is in reality Ac?

Our mistake, It is Ac-VRPR. We have corrected it in the manuscript (line 442 in the manuscript87 below).

4. Along the same thought, the gene expression measurement is very important, but I
must say, that does not mean that the enzymatic activity you are measuring corresponds
to the expressed gene, if, as said before, that specific activity has not been measured in a
purified protein. Hence, caution is needed on this respect when interpreting your data.

We certainly agree. We do measure metacaspases gene expression, but we do not know if the
enzymatic activity we are measuring corresponds to the expressed gene. In our previous
experiment in the New Caledonian lagoon (Spungin et al., 2016) we measured MC gene expression
via metatranscriptomics during different stages of bloom demise. Also, in our submitted manuscript (Spungin et al., in review EM) we measured MC gene expression by applying qRT-PCR 98 for both field and cultures. We found that MC gene expression is highly elevated during different 99 stages of bloom demise / PCD induction. While we are just beginning to elucidate the roles of the 100 different metacaspases (12 in *Trichodesmium*) we still cannot directly link between expression and 101 activity. Here, in this manuscript we did not specifically examine the MC gene expression as we 102 have previously demonstrated higher expression of metacaspase during bloom demise and PCD 103 induction (Spungin et al. 2016, Bar Zeev et al. 2013). In this study we focused on activity of the 104 metacaspase and caspase-like (as measured by specific substrates) proteins as potential PCD 105 markers. Yet, as the reviewer notes, we do not know what specific protein is responsible for the 106 caspase-like activities and what drivers regulate it, thus it cannot be directly linked to gene 107 expression. We have clarified this in the text (lines 692-693 in the manuscript below)

5. Additionally, I think we all must accept that we do not really know if there are initiator 110 or executor CLs or MCs in phytoplankton. Two types of metacaspases (types I and II) are 111 defined based on the presence of a prodomain analogous to the classification of caspases 112 into initiator or executioner caspases. The molecular role of a prodomain in initiator 113 caspases is the recruitment of caspases to multicomponent signaling complexes for 114 caspase activation. However, phytoplankton metacaspases often lack prodomains (Choi 115 and Berges 2013). As I see it, to use this homology can lead into mistake, so I would not 116 describe the enzymes involved as executors of the cell, or initiators of the cascade 117 (although for vascular plants is widely used, it is different, they know exactly which 118 protease which is, and what they do).

- Thank you for this valid clarification. We have deleted these suggestions from the text, so we donot define these enzymes as executors or initiators (lines 368-369 in the manuscript below).
- 121

6. Last but not least, just would like to know your opinion on this actual heated-debate:
Do you think that at the time being caspase-like proteins, in phytoplankton, could
hydrolyze R or V?

We have extensively worked on metacaspases vs caspases-like proteins trying to elucidate the 126 differences/ common roles of both in *Trichodesmium* in lab and field extracts (Spungin et al., in 127 review EM). As our experiments find a significant positive correlation between both activates, we 128 have done a series of inhibitor experiments. In vitro treatment with a metacaspase inhibitor-129 antipain dihydrochloride, efficiently inhibited metacaspase activity, confirming the arginine-based 130 specificity of *Trichodesmium* metacaspases (see Fig. 1). Our biochemical activity and inhibitor 131 observations demonstrate that metacaspases and caspases-like activities are likely distinct and 132 are independently activated under stress and coupled to PCD in our experiments of both 133 laboratory and field populations. However, caspase-like activity was somewhat sensitive to the 134 metacaspase inhibitor, antipain, showing a ~30-40% drop in activity. This hints at some catalytic 135 crossover between these two catalytic activities in *Trichodesmium* that further should be studied. 136 We have also inserted this issue to the discussion in this manuscript (lines 741-751 in the manuscript below).

(Fig. 1 Spungin et al., in review EM)

**7.** The second major concern relates to the fact of bloom/ cell dismissal in the water column.

When working in the field, dead cells are rarely seen at later stages (Berges and Choi 2014) 156 or not seen (Segovia et al., 2018), only because they have been cleared away from the system. Any source of energy that cellular debris may provide to the neighborhood will be 157 158 immediately used by other species within the food web. So, it is very unlikely to see cellular 159 rests consequence of CD on the water column. Yet, POC downward flux is the way to have 160 some estimates. In my opinion and experience, this can be applied to cultures in the lab 161 under controlled conditions, but I find it truly complicated in natural communities / 162 ecosystem level. Please, clarify how this fits within your sampling/sample analyses time framework. Has that to do with the blooming condition excluding other components of 163 164 the trophic web of the niche?

The assumption that most dying and dead cells are utilized quickly and recycled within the food 166 web and upper surface layer, may be correct especially in the surface layers of the oligotrophic 167 oceanic regions. Yet, when high biomass blooms occur (as with Trichodesmium blooms) the fate of 168 the extensive biomass is more complicated (Bonnet et al., 2015). PCD induced cell death, combined 169 with buoyancy loss, can lead to rapid sinking to depth of the biomass at a speed that would prevent 170 large feeding events on this biomass. This may be determined by POC downward fluxes easy to 171 measure in the lab and extremely complex in the open ocean as you mentioned. We previously 172 measured POC export in our lab under controlled conditions (Bar-Zeev et al., 2013). In this specific 173 experiment however, as mentioned in the text, we had also deployed sediment traps (150, 325, 174 and 500 m depth). In these sediment traps we measured POC fluxes, but also have specific 175 indications (NifH reads) of Trichodesmium and other diazotrophs which were blooming for several 176 days at the surface. This indicates that under bloom conditions when biomass is high some of the 177 cell pellets do sink down out of the food web. This has also been added and discussed in the text 178 (lines 880-891 in the manuscript below).

8. Nothing is said about viruses affecting C losses, which is important for C cycling and
definitively affects C export. Viruses were not measured the text says. But in my opinion,
this shall at least be discussed and do not directly exclude this possibility as a possible
cause for bloom demise. Is there any long-term study done on Trichodesmium blooms
termination affected by viruses that at least allows you to compare with other situations?

Viruses have been increasingly invoked as key agents terminating phytoplankton blooms. Infection 185 by phages has been invoked as the mechanism of Trichodesmium bloom crashes, (Brown et al., 186 2013; Hewson et al., 2004; Ohki, 1999) but it has yet to be unequivocally demonstrated in long 187 term *Trichodesmium* blooms. We did study this in a natural bloom of *Trichodesmium* in the new 188 Caledonian lagoon. Virus like particles were measured from samples collected from the bloom 189 during different stages of demise. Enumeration of virus-like particle numbers did not indicate that 190 a massive, phage-induced lytic event of Trichodesmium occurred there. This issue was discussed 191 and published in Spungin et al., 2016 Bigeoscience. 192 As Trichodesmium spp. are not grazed by predominant copepods of the water column because of

As *Trichodesmium* spp. are not grazed by predominant copepods of the water column because of 193 toxins, we believe that PCD and particularly viral lysis may be considerable sources of mortality.

Virus infection may also induce PCD in *Trichodesmium*: Virus infection has been shown to increase

- the cellular production of reactive oxygen species (Vardi et al., 2012), which in turn can stimulate
  PCD in algal cells (Berman-Frank et al., 2004; Bidle, 2015; Thamatrakoln et al., 2012). Viral attack
- can also directly trigger PCD as part of an antiviral defense system activated to limit virusproduction and prevent massive viral infection (Bidle, 2015).
- 199 We have now mentioned and discussed this further in the text (lines 659-668 in the manuscript200 below).

**202**

- Spungin, D., Bidle, K.D and Berman-Frank, I. The roles of Metacaspases in programmed cell death
   of the marine cyanobacterium *Trichodesmium*. Environmental microbiology, In review.

[revised manuscript text omitted]
 | Depth        | ТЕР                      | TEP-C | %TEP-C | POC  | TOC  | POC/PON |
|---------|----|--------------|--------------------------|-------|--------|------|------|---------|
| LDA     |    | ( m ) | (µg GX L -1 ) |       |        | (µM) | (µM) |         |
| station |    |              |                          |       |        |      |      |         |
| 1       |    | 200          | 296±135                  | 186.5 | 27.2   | 3.04 | 57.2 | 5       |
|         |    | 150          | ND                       | ND    | ND     | 3.18 | 61.1 | 13      |
|         |    | 70           | 87±17                    | 54.8  | 6.7    | 2.93 | 68.7 | 11      |
|         |    | 50           | 562±7                    | 354.3 | 41.9   | 2.47 | 70.5 | 13      |
|         |    | 10           | 241±40                   | 152.3 | 14.5   | 9.21 | 87.4 | 8       |
| 3       |    | 200          | 191±13                   | 120.9 | 18.6   | 1.29 | 54.2 | 27      |
|         |    | 150          | $144 \pm 54$             | 91.2  | 12.9   | 2.22 | 59.0 | 22      |
|         |    | 80           | 263                      | 166.1 | 20.5   | 4.62 | 67.5 | 15      |
|         |    | 10           | 126±2                    | 79.6  | 8.3    | 3.60 | 79.7 | 12      |
| 5       |    | 200          | 200                      | 126   | 21.3   | 2.84 | 54.2 | 236     |
|         |    | 150          | 220                      | 138.6 | 18.0   | 2.72 | 58.2 | 7       |
|         |    | 80           | 146                      | 92.2  | 12.1   | 4.91 | 63.3 | 8       |
|         |    | 50           | 348±60                   | 219.5 | 26.8   | 3.33 | 68.3 | 6       |
|         |    | 10           | ND                       | ND    | ND     | 5 80 | 837  | 7       |

**Table 2-** Temporal changes in the relative composition (w/w) and distribution of TEP, TEP-C and
organic carbon and nitrogen fractions within the water column during days 1,3 and 5 in the LDB
station at different depth ranging between surface (7 m) to 200 m.

| Day     | at | Depth        | TEP                      | TEP-C | %TEP-C | POC  | TOC  | POC/PON |
|---------|----|--------------|--------------------------|-------|--------|------|------|---------|
| LDB     |    | ( m ) | (µg GX L -1 ) |       |        | (µM) | (µM) |         |
| station |    |              |                          |       |        |      |      |         |
| 1       |    | 7            | 408±36                   | 257.1 | 23.4   | 8.95 | 91.5 | 6.0     |
|         |    | 35           | 279±86                   | 175.9 | 17.0   | 5.86 | 86.0 | 9.1     |
|         |    | 100          | 214±67                   | 134.7 | 16.8   | ND   | 66.7 | ND      |
|         |    | 150          | 145±34                   | 91.5  | 12.3   | 3.79 | 61.9 | 11.2    |
|         |    | 200          | 244±113                  | 153.7 | 20.3   | 7.61 | 63.2 | 9.8     |
| 3       |    | 7            | 402±12                   | 253.1 | 22.5   | 8.88 | 93.9 | 6.9     |
|         |    | 35           | 193±48                   | 121.8 | 12.6   | 3.07 | 80.3 | 8.2     |
|         |    | 100          | 163±33                   | 102.4 | 12.6   | ND   | 67.8 | ND      |
|         |    | 150          | 145±34                   | 91.6  | 12.0   | 1.91 | 63.8 | 7.4     |
|         |    | 200          | 127±79                   | 80.2  | 11.3   | 1.71 | 59.3 | 5.7     |
| 5       |    | 7            | 565±87                   | 355.8 | 32.5   | 5.32 | 91.3 | 5.9     |
|         |    | 70           | 294±53                   | 185.2 | 20.1   | 2.21 | 76.7 | 6.1     |
|         |    | 100          | 264±160                  | 166.2 | 19.6   | 2.25 | 70.6 | 8.0     |
|         |    | 150          | 224±51                   | 140.8 | 15.9   | 1.53 | 73.9 | 5.1     |
|         |    | 200          | 231±45                   | 145.8 | 21.1   | 1.11 | 57.6 | 5.5     |

Abbreviations: TEP, transparent exopolymeric particle; TEP-C, TEP carbon; POC, particulate
organic C; TOC, total organic C; ND- no data.

Figures

Figure 1

---

## Author Response (AR2)

**We would like to thank again referee #2 for his insightful comments for clarification and improvement of the manuscript. We thank your approval of the paper and have addressed the minor corrections as requested. Our answers follow the comments in black (bold). A marked-up manuscript version with the relevant changes made is also included.**

**1.** There are still places were the word caspase has not been substituted by caspase-like (some figure axis etc and in the text).

**We have now replaced all the remaining caspase activities to caspase-like activities (specifically in the text lines- 173,179,189,480). Regarding the figure axis- we do not use the term caspase-like activity in the figure axis, but as requested it appears as IETDase cleavage, which is the right term.**

**2.** The authors should explicit that the metacaspase activities observed do not correspond to a purified protein, but rather to a cell free extract, and that this is just a limitation of the method 
[revised manuscript text omitted]

[Figure]

**Figure 2**

[Figure]

**Figure 3**

[Figure]

**Figure 4**

[Figure]

**Figure 5**

[Figure]